# Neurons exploit stochastic growth to rapidly and economically build dense dendritic arbors

Xiaoyi Ouyang[1,2,7], Sabyasachi Sutradhar [1,7], Olivier Trottier [1,2,6], Sonal Shree[1], Qiwei Yu [3], Yuhai Tu[4] & Jonathon Howard [1,2,5] ✉

Dendrites grow by stochastic branching, elongation, and retraction. A key question is whether such a mechanism is sufficient to form highly branched dendritic morphologies. Alternatively, does dendrite geometry depend on signals from other cells or from the topological hierarchy of the growing network? To answer these questions, we developed an isotropic and homogenous mean-field model in which branch dynamics depends only on average lengths and densities: that is, without external influence. Branching was modeled as density-dependent nucleation so that no tree structures or network topology was present. Despite its simplicity, the model predicted several key morphological properties of class IV Drosophila sensory dendrites, including the exponential distribution of branch lengths, the parabolic scaling between dendrite number and length densities, the tight spacing of the dendritic meshwork (which required minimal total branch length), and the radial orientation of branches. Stochastic growth also accelerated the overall expansion rate of the arbor. We show that stochastic dynamics is an economical and rapid space-filling mechanism for building dendritic arbors without external guidance or hierarchical branching mechanisms. Our work therefore provides a general theoretical framework for understanding how macroscopic branching patterns emerge from microscopic dynamics.

Neurons have the most complex morphologies of all cells. In the fly brain alone, over 8000 neuronal types are distinguished, primarily by their shapes[1,2]. Different shapes support different functions, which is beautifully illustrated in the cases of mammalian retinal ganglion cells[3] and fly dendritic arborization (da) cells[4]. Much of the diversity in shape arises from the dendrites, which are branched processes that extend from the cell bodies of neurons and receive signals from other neurons or the environment. To receive many inputs, dendrites often form highly branched arbors whose large surface areas can accommodate many post-synaptic sites or sensory receptors[5]. Branching is important because it decreases the distance from the synapses or receptors to the cell body, thereby reducing propagation times and signal losses compared to unbranched geometries[6]. Propagation times are also reduced by the often-radial orientation of the branches so that distal signals more quickly reach the centrally located cell body. Branch points serve as intermediate sites of integration and computation of the information encoded spatially within the arbor[7,8]. In these ways, the branched geometries of dendrites are key to neuronal function.

[1]Department of Molecular Biophysics and Biochemistry, Yale University, New Haven, CT 06511, USA. [2]Department of Physics, Yale University, New Haven, CT 06511, USA. [3]Lewis-Sigler Institute for Integrative Genomics, Princeton University, Princeton, NJ 08544, USA. [4]Center for Computational Biology, Flatiron Institute, New York, NY 10010, USA. [5]Quantitative Biology Institute, Yale University, New Haven, CT 06511, USA. [6]Present address: Department of Chemical and Physical Sciences, University of Toronto - Mississauga, Toronto, ON M5S 1A1, Canada. [7]These authors contributed equally: Xiaoyi Ouyang, Sabyasachi Sutradhar. ✉e-mail: joe.howard@yale.edu

A fundamental question is: how do branched arbors form and grow? A partial answer comes from live-cell imaging, which shows that dendrite growth is highly dynamic. During neuronal development and regeneration, new branches form by lateral branching from existing branches, and the growing tips of dendrites transition between growing and shrinking states: frog brain[9,10], cultured[11] and acute[12] brain slices, retinal explants[13], mouse brain[14], cultured neurons[15], living flies[16] and worms[17]. Importantly, transitions from growing to shrinking states can occur either spontaneously or after the collision of the growing tip with another dendrite, mediated by self-avoidance molecules[18,19]. In fly da neurons, where dendrite morphogenesis has been studied in detail, the dynamics is highly stochastic[20]: branching occurs throughout the arbor, and the transitions between growing and shrinking states are random, like the dynamic instability of microtubules[21].

Can the stochastic dynamics of dendrite tips generate the observed dendritic arbors of neurons? So-called agent-based models, which simulate the branching and growth of each tip individually, give a partial answer to this question. They can generate several of the observed features of dendritic arbors in Purkinje cells[15], *Drosophila* class I cells[22,23], *Drosophila* class IV cells[20], and other neurons[24–27]. An advantage of agent-based models is that they are bottom-up and so mimic developmental processes. This distinguishes them from top-down models that follow a predetermined template[6]. Agent-based models have several limitations, however. First, they are complicated and may depend on the details of the simulations[28]. Second, the complexity makes it difficult to relate the large-scale predictions of the simulations, such as branch lengths and densities, to the small-scale behavior: for example, how does branch length depend on branching and growth rates? This will be important when relating molecular and genetic processes underlying branching and growth to the phenotypic properties of morphologies[4,19]. Third, there are other mechanisms not addressed in agent-based models. For example, two-step models distinguish between "main" branches and peripheral branches, and have successfully recapitulated the morphologies of *Drosophila* class III cells, whose branches are decorated with branchlets that depend on the actin cytoskeleton[29]. Other models consider dendrites as a hierarchy of primary, secondary, tertiary, etc. branches and propose that the formation of new branches depends on the position within the hierarchy[22,30,31]. In other words, growth may depend on the topology of the developing dendrite, which cannot be tested by agent-based models because they have their own inherent topology.

In this work we address the limitations of agent-based models, by developing a "mean-field" model to test whether stochastic dynamics alone can generate observed dendrite morphologies. Mean-field models have a rich history in physics and have successfully been applied to several areas of biology, including electrical excitability[32], patterning[33], and collective migration[34]. The advantage of mean-field models over agent-based ones is that they greatly reduce the number of variables. For example, even though electrical excitability can be simulated by an agent-based model with thousands of individual ion channels, it is the mean-field theory, which takes the form of the Hodgkin-Huxley equation, that revealed the dependence of the speed of the action potential on the rate and voltage-dependence of channel gating, axon diameter, and the passive electrical properties of the membrane[35]. In the case of dendrites, the number of degrees of freedom is reduced from the thousands of individual branches to just the branch number density and the branch length density. This simplification allows analytic solutions to global morphological properties such as the statistics of branch lengths and densities in terms of the microscopic properties such as the branching rates, tip speeds, etc. Furthermore, because branching is replaced by nucleation, the mean-field approach does not simulate arbors nor consider the tree structure of the networks. Therefore, the field does not have topology. Yet, as we show, the model can quantitatively account for many geometric properties of *Drosophila* class IV cells, including branch length distributions, branch densities and orientation, the rate of arbor expansions, and several empirical scaling laws. Thus, mean-field models can test whether topology is necessary for arbor geometry. We discuss how the model could provide insight into mutations and how the model, which is formulated for two-dimensional arbors, might generalize to other neurons, including those with 3D arbors.

## Results

### Class IV da neurons form a dense network of dendritic branches

We used fly sensory neurons to investigate the relationship between the dynamics of dendrite branches and arbor morphology. The highly branched arbors of *Drosophila* class IV da nociceptors sense noxious stimuli, including heat[36], ultra-violet light[37], and harsh mechanical stimuli[38] over a large area of the larval surface (Fig. 1a, b)[4,20]. Because the dendrites grow on the approximately planar surface of an extra-cellular matrix[39], the arbor is quasi-two-dimensional: this simplifies image processing and theoretical analysis. By 48 h after egg-lay (AEL), the arbors fill the larval segments with a fine meshwork of dendrites (called "tiling"[40]), which serves to detect acute localized stimuli such as penetration of the cuticle by the ovipositor barbs of parasitoid wasps[41] (Fig. 1a). Electrical signals, initiated by cell-surface receptors, are conducted to the cell body before being relayed to the central nervous system[42,43] where they initiate escape responses, such as rolling, to displace the wasp before it lays an egg[38]. While these dendrites do not receive synaptic input, their morphology[19] and topology[44] resemble those of central neurons such as retinal ganglion cells and Purkinje cells, which are also approximately planar. Therefore, principles learned from class IV cells may provide insight into other 2D arbors (and see Discussion for how the principles may extend to 3D arbors).

Over the 5 days of development, class IV arbors grow from 100 μm to 500 μm in diameter[40] (Fig. 1c–e). The length density of dendrites, defined as the total branch length per unit area, reaches a plateau in the central region (uniform color in Fig. 1c–e), suggesting that a steady state has been reached there. The density then spreads as a moving front (Fig. 1f, g). The plateau levels decrease by about 50% over development while the widths increase 5-fold.

There are two types of branches (Fig. 2a, b): terminal branches extend from a branch point to a tip, and internal branches connect either two branch points or a branch point and the soma. The number of terminal branches is approximately equal to the number of internal branches (Fig. 2a). The average lengths of terminal and internal branches are also similar (Fig. 2c). Both types of branches have exponential length distributions, with similar means at all developmental stages (Fig. 2d, Supplementary Fig. 1a–c).

Dendrite branches tend to be radially oriented: the radial angle ($\theta$ in Fig. 2a) that each branch makes with the radial direction (example in Fig. 2e) is peaked in the outward direction (Fig. 2f–h, Supplementary Fig. 1d–f). Further analysis revealed that internal branches predominantly contribute to this peak (Fig. 2h), whereas terminal branches are more isotropic (Fig. 2g). In this work, we show that these properties are a consequence of the dynamics of the dendrite tips of these cells.

### Branching, growth, retraction, and branch interconversion occur on short timescales

The dendrite tips of class IV cells are highly dynamic on the minute timescale. As previously reported by Shree et al. [20], after the formation of a new branch (which occurs on both terminal and internal dendrites), tips grow, shrink, and pause, and following collision with the shaft of another dendrite, they retract and disappear (Fig. 3a, b). In addition to these behaviors, we discovered that branches sometimes reappear at the sites of disappearance (Fig. 3c). We measured the probability of reappearance, $\beta$, which we call the rebranching

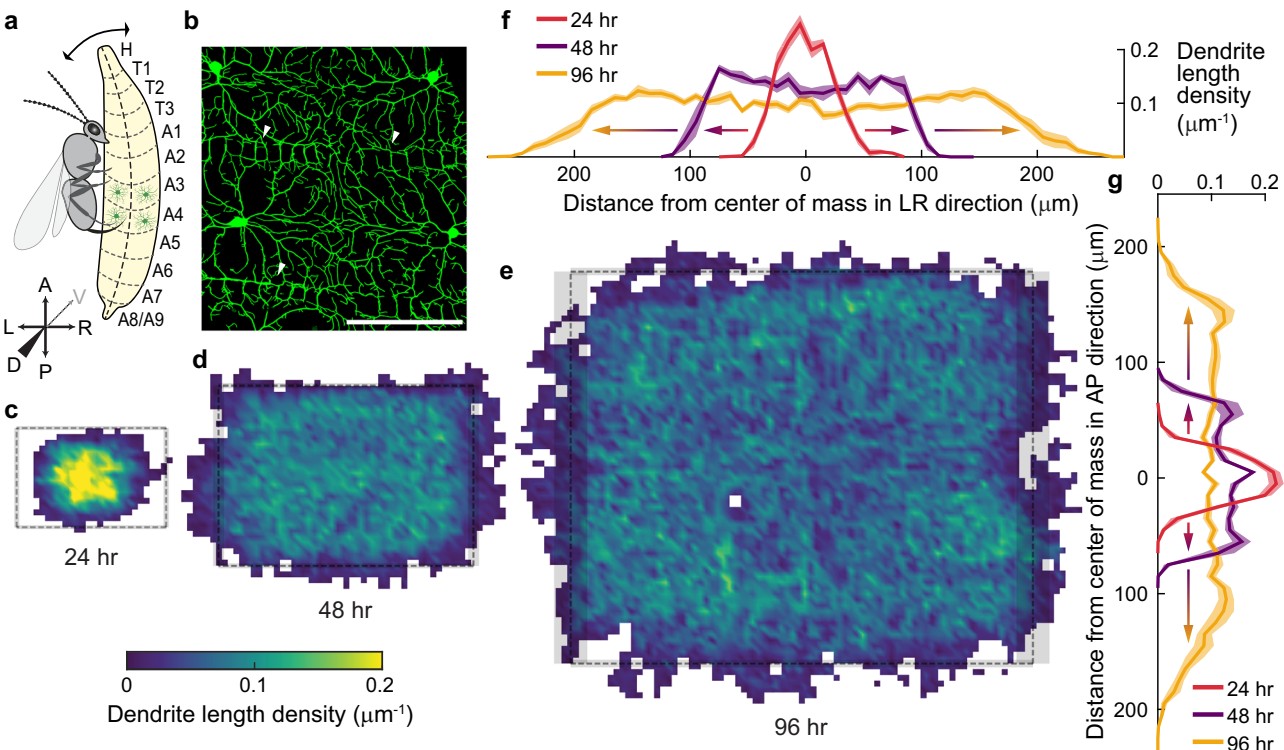

**Fig. 1 | Growth of class IV neurons during development. a** Schematic of a *Drosophila* larva from the dorsal side attacked by a parasitoid wasp. Larval body plan axes are marked with anterior (A), posterior (P), dorsal (D), ventral (V), dorsal left (L), and dorsal right (R). The size of the larva is exaggerated in comparison to the wasp. **b** Class IV neurons (48 h after egg-lay, AEL) are marked with the transmembrane protein CD4 tagged with green fluorescent protein (GFP) (genotype is;;ppkCD4-tdGFP) and imaged using a 40x water immersion objective by spinning-disk confocal microscopy and displayed as the maximum projection of 10 sections (see Methods). The scale bar is 100 μm and also applies to (**c**–**e**). White arrowheads indicate tilings. **c**-**e** Coarse-grained dendrite length density of class IV neurons at 24 (13 cells from 10 animals), 48 (12 cells from 12 animals), and 96 (9 cells from 9 animals) hours AEL, respectively. The rectangles represent the corresponding average segment sizes at each developmental stage, with gray-shaded regions being the standard error of the mean (9 cells from 3 animals for each developmental stage). **f, g** Mean dendrite length density along the AP and LR axes at different developmental stages: 24 h in red, 48 h in purple, and 96 h in yellow. The scales in (**f**, **g**) are the same as for (**b-e**).

probability, and found it to be ~0.2 throughout development (Fig. 3d). The rate of rebranching is much larger than expected for de novo branching (Fig. 3e, see "Experiments and Image Analysis" in Methods for details). Rebranching is important conceptually, and it improves the quantitative agreement between the model and the data. The rates of branching, growth and shrinkage, and the transitions between states, together with rebranching, are the input to the mean-field model (Table 1).

Branching and debranching can create and destroy terminal and internal branches. When a new branch forms on a terminal branch, the proximal part of the terminal branch becomes a new internal branch (Fig. 3b and see Supplementary Fig. 2a). Conversely, when one of the sibling terminal branches disappears, the parental (internal) branch fuses with the sibling to become a longer terminal branch (Fig. 3b(iii) and Supplementary Fig. 2b, c). In these cases, terminal and internal branches interconvert. In contrast, branching and debranching on an internal branch do not interconvert branch types (Supplementary Fig. 2d–f). These are the rules by which terminal and internal branches are created and lost. The net result is that the numbers and length distributions of internal and terminal branches are equal (Fig. 2c, d and Supplementary Eq. (8)).

### Three-state mean-field model

In this section, we describe the mean-field model in non-mathematical language. The equations are formulated in the next section and solved in the Methods. In the mean-field model, every position in the arbor is associated with four properties: average branch number density

($N$, branches per unit area), average branch length density ($\rho$, the branch length per unit area), the distribution of branch lengths, and the distribution of the radial orientation of the branches (relative to the radial angle). The branches can be growing, shrinking, or paused and switch stochastically between these states, analogous to the dynamical instability of microtubules[45]. We initially focus on terminal branches. Internal branches are accounted for quite simply because they have similar numbers and lengths to terminal branches (Fig. 2c, d and Supplementary Information: "Full Model" section). This mean-field approach is a great simplification over agent-based models, which keep track of the hierarchical order of thousands of growing and interacting branches.

Branching is implemented as nucleation proportional to the local dendrite length density. This mimics random branching along existing branches, consistent with spatially uniform branching throughout the arbor[20]. This leads to spatial homogeneity in the central region of the arbor (Fig. 4a). By modeling branching as nucleation, the arbor can be thought of as a set of directed lines (Fig. 4b,c) with no topology. Branching is assumed to be isotropic, without directional preference, similar to the observed broad distribution of new branches[20]. This leads to isotropic terminal branches in the central region. Branches are born in the growing state, and then stochastically transition between the three states (growing, shrinking and paused) according to the measured rates (Table 1). The branches are assumed to be straight, consistent with their high persistence length[20]. In our implementation of the model, there are no external cues that could lead to inhomogeneities or anisotropies.

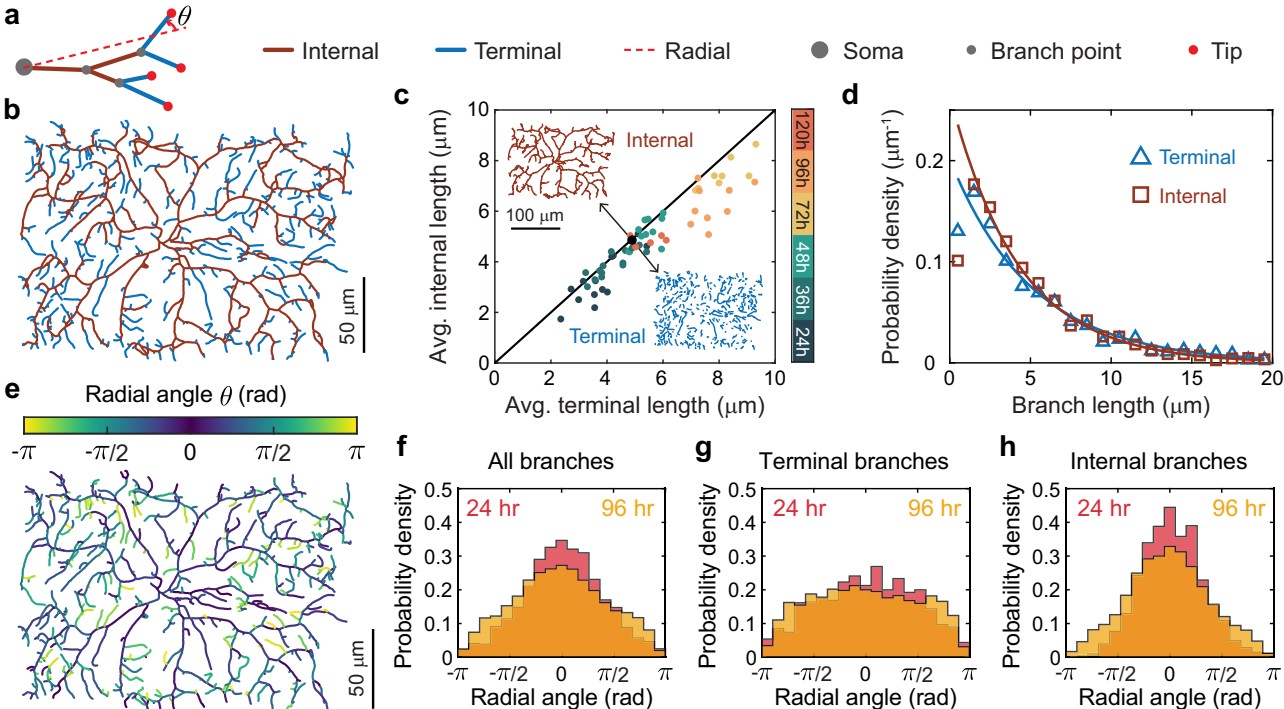

**Fig. 2 | Distribution of branch length and orientation. a** The schematic illustrates branch classification. A terminal branch (blue segment) extends from a branch point (small gray circle) to a tip (red circle), while an internal branch (brown segment) connects either two branch points or a branch point and the soma (large gray circle). For a binary tree, the number of terminal branches exceeds the number of internal branches by one. Theta ($\theta$) denotes the angle of the branch relative to the radial direction (red dashed line). **b** The skeleton of an example class IV neuron at 48 h AEL showing terminal branches in blue and internal branches in brown. **c** The average lengths of terminal and internal branches are similar (61 cells ranging from 24–120 h AEL). The black line has a slope of one. The black point corresponds to the cell in (**b**). Insets show the terminal and internal branches of the dendrite in (**b**). **d** Branch lengths at 48 h AEL are shown with exponential fits (ignoring the first bin). **e** The branches in the same cell as (**b**) are color-coded by radial angle. **f**–**h** Radial-angle distributions for all branches, terminal branches only, and internal branches only at 24 and 96 h AEL. The distributions are peaked in the outward direction in (**f**) and (**h**) but remain mostly flat in (**g**).

Collision is modeled as the instantaneous elimination of growing dendrites following contact with pre-existing branches. This is motivated by the observation that, following collision, the dynamics is strongly biased towards the shrinking state[20]. The collision rate increases with the average growth speed, $\bar{v}$, as in a chemical reaction, and also increases due to the length fluctuations caused by stochastic growth (a positive fluctuation may cause a collision sooner than if the growth were constant). This latter contribution is proportional to the diffusion coefficient $D$ associated with the stochastic growth[45] (also see Methods: "Drift and diffusion").

Although this coarse-grained approach overlooks the fine structure and topology, it is amenable to mathematical analysis and has predictive power. In parallel, we implemented a directed-rod simulation that mimics the mean-field representation (Fig. 4c, Supplementary Fig. 3a, Methods: "Directed-rod simulation" section), which provides a visualization of the mean-field model, allows us to determine collision-associated geometric pre-factors, and serves as a check on the analytic solutions.

### Mathematical formulation of the three-state model

The mean-field equations for the dendrite densities $n_X(r, l, \theta, t)$ for terminal branches with length $l$, with radial angle $\theta$, at radial position $r$, at time $t$, in state $X \in (G, S, P)$ are:

$$\frac{\partial n_G(r, l, \theta, t)}{\partial t} = -(k_{GS} + k_{GP})n_G + k_{SG}n_S + k_{PG}n_P - v_G \frac{\partial n_G}{\partial l} - K_{col}(r, t)n_G - \frac{v_G}{2}\mathcal{R}(r, \theta)n_G \tag{1}$$

$$\frac{\partial n_S(r, l, \theta, t)}{\partial t} = k_{GS}n_G - (k_{SG} + k_{SP})n_S + k_{PS}n_P + v_S \frac{\partial n_S}{\partial l} + \frac{v_S}{2}\mathcal{R}(r, \theta)n_S \tag{2}$$

$$\frac{\partial n_P(r, l, \theta, t)}{\partial t} = k_{GP}n_G + k_{SP}n_S - (k_{PG} + k_{PS})n_P \tag{3}$$

See parameter descriptions in Table 1. The 0th and 1st order moments of the dendrite density $n_X(r, l, \theta, t)$ in state $X \in (G, S, P)$ with respect to $l$ and integrated over $\theta$ are:

$$N_X(r, t) \equiv \int_{-\pi}^{\pi} d\theta \int_0^{\infty} dl\, n_X(r, l, \theta, t) \tag{4}$$

$$\rho_X(r, t) \equiv \int_{-\pi}^{\pi} d\theta \int_0^{\infty} dl\, l \cdot n_X(r, l, \theta, t) \tag{5}$$

We call $N_X(t)$ the number density (number of dendrites per unit area) in state $X$ and call $\rho_X(t)$ the length density (dendrite length per unit area) in state $X$. In addition, we let $N_T \equiv \sum_{X \in (G, S, P)} N_X$ and $\rho_T \equiv \sum_{X \in (G, S, P)} \rho_X$ be the terminal branch number and length density, respectively. The total number and length density including internal branches are adjusted by doubling those of terminal branches, yielding $N_{tot} = 2N_T$ and $\rho_{tot} = 2\rho_T$. See Supplementary Information for formal derivations using a full model with terminal-internal branch interconversions. The average branch length equals $\bar{l} = \rho_{tot}/N_{tot}$.

The three left-most terms in Eqs. (1-3) are the tip transitions between the three states, with rates given in Table 1. The transitions are assumed to be Markovian, for which there is experimental support[20].

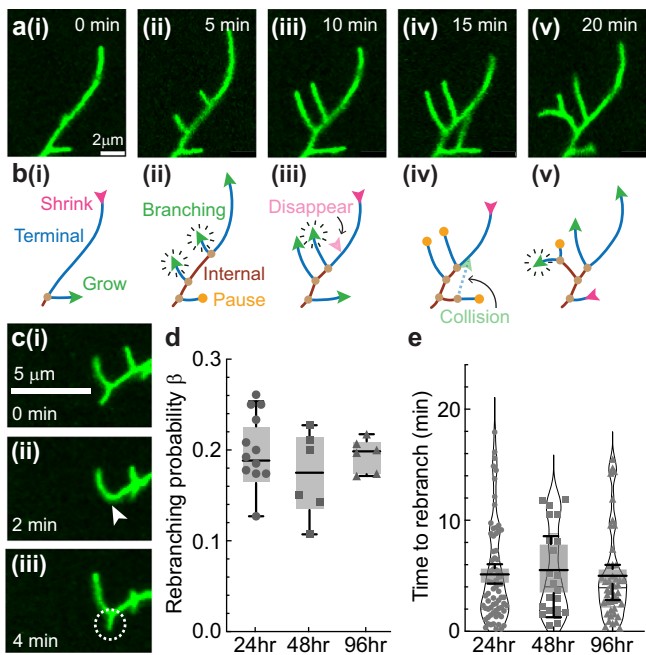

**Fig. 3 | Dendrite dynamics. a** Time-lapse images illustrating the microscopic behavior of dendrite branches at 24 h AEL. **b** Schematics of the branch dynamics in (**a**). Growing, shrinking, and paused tips are indicated by green, magenta, and orange arrowheads, respectively. The green arrowheads with radiating lines indicate branching events. The faded green arrowhead with a dashed line indicates a collision. The faded magenta arrowhead indicates spontaneous branch disappearance. **c** Time-lapse images of a 24 h class IV dendrite tip showing a debranching event (white arrowhead indicates the site of tip disappearance) and the subsequent rebranching (white circle). **d** The probability of rebranching within 1520 min of debranching at different developmental stages (24 h: 12 cells from 6 larvae, 48 h: 6 cells from 6 larvae, 96 h: 6 cells from 6 larvae). The points represent cells, and the box-and-whisker plots represent animal averages: long horizontal lines are medians, the boxes enclose the second and third quartiles, and the short horizontal lines (whiskers) show the range. If rebranching were simply due to spontaneous branching near the debranching location (within ± 2 pixels), then the probability of a spontaneous branch in a 15 min recording time would be 0.07 (24 h, 106 nm pixel), 0.021 (48 h, 162 nm pixel) and 0.012 (96 h, 162 nm pixel), smaller than the observed probabilities. The differences between ages are not statistically significant (one-way ANOVA, Tukey correction, 5% level). **e** Time to rebranch at different developmental stages. (24 h: 71 rebranching events from 6 larvae, 48 h: 22 events from 6 larvae, 96 h: 44 events from 6 larvae). Box-and-whisker plots are defined as in (**d**). Spontaneous branching would lead to apparent rebranching times of 300 min, 1000 min, and 1700 min respectively, much longer than the observed times. The differences between ages are not statistically significant (one-way ANOVA, Tukey correction, 5% level).

The fourth term $-v_G \partial n_G / \partial l$ in Eq. (1) and $v_S \partial n_S / \partial l$ in Eq. (2) describe how elongation or retraction physically shifts the dendrite densities at a certain length. These terms are an extension of the Dogterom-Leibler model of microtubule dynamic instability[45] to three states.

The term $K_{col}(r, t)n_G$ in Eq. (1) is an empirical collision term. Using directed-rod simulations, we found the collision rate to be $K_{col}(t) = [\alpha \bar{v}\rho(t) + \alpha^2 \gamma D\rho^2(t)]N_T(t)/N_G(t)$, where the first and second term represents the advective and diffusive component of collision (Methods: Fitting of collision parameters). Here, $\alpha$ and $\gamma$ are two empirically determined dimensionless pre-factors that depend weakly on the dynamic parameters. Note that $N_T(r, t)/N_G(r, t)$ is a normalization factor to ensure all collisions occur in the growing state.

The terms with the transport operator $\mathcal{R}(r, \theta) \equiv \cos\theta\partial_r - \sin\theta(1/r)\partial_\theta$ in Eqs. (1) and (2) correspond to changes in radial position and orientation due to branch elongation and retraction. A factor of ½ appears because the branch midpoints, where position and orientation are measured, move at half the speed of the tips.

The boundary condition at length zero ($l = 0$) is:

$$k_b\rho(r,t) = \int_{-\pi}^{\pi} d\theta \left[ v_G n_G(r, l = 0, \theta, t) - \beta v_S n_S(r, l = 0, \theta, t) \right] \quad (6)$$

The left-hand side is the rate of increase of the number of new dendrites per unit area due to branching nucleation with rate $k_b$ (per unit length per unit time). The right-hand side is the rate at which these newly formed dendrites move away from zero length less a rebranching term, which effectively increases nucleation when a fraction $\beta \cong 0.2$ (Fig. 3d) of shrinking branches switch to growing.

### One-state model

Before describing the predictions of the three-state model, we first present the results of a simpler one-state model where branches grow with a constant velocity $\bar{v}$ without pause or shrinkage, and branches disappear only by collision (Methods: "Steady-state solution of the one-state model"). This corresponds to the limit where the state transitions occur at high frequency. The solution (Table 2) shows that the macroscopic mean-field properties of the arbor−number density ($N$), length density ($\rho$), and mean length ($\bar{l}$)−can be expressed simply in terms of the microscopic properties of the tips−the branching rate ($k_b$), the velocity ($\bar{v}$), and the collision frequency ($\alpha$). For example, at steady state, the branch birth rate per unit time ($k_b\bar{l}$) is equal to the branch death rate ($\bar{v}/\bar{l}$) due to collisions; therefore, the mean length is $\bar{l} \propto \sqrt{\bar{v}/k_b}$. The predictions from the one-state model, however, do not agree quantitatively with the data (red circles in Fig. 5a), showing that stochastic transitions between the three states are, therefore, important.

### The three-state model predicts the observed branch lengths and densities in the center of the arbor

To compare the model with the data, we obtained steady-state solutions to the three-state model using the measured microscopic parameters in Table 1. Both numerical (Supplementary Fig. 4) and analytic (Eq. (14)) solutions show that the steady state is reached in only a few hours, being limited by branching, which has the slowest rate in the system. Because this time is much shorter than the days-long developmental time, the central region can be regarded as being in a quasi-steady state throughout development. With internal branches incorporated, the mean-field model predicts that the length distributions are exponential, as observed (Fig. 2d). The exponential distribution is a consequence of tip dynamics and collisions being independent of branch length, so that branch survival is a Poisson process. The model predicts the measured average dendrite lengths and length densities at all developmental stages (Fig. 5a,b); the agreement is improved when the rebranching is included. These agreements show that stochastic dynamics accounts for the average densities in the central plateau region.

### The model predicts the parabolic relationship between number and length densities

We found that, throughout development, the number density ($N$) is approximately proportional to the square of the length density ($\rho$) in the central region (Fig. 5c). This approximately parabolic relationship between $N$ and $\rho$ is predicted by the mean-field model (Fig. 5c, solid curve, Methods: "Parabolic relation") using the parameters in Table 1. The one-state model predicts an exact parabola. The parabolic relation was first found by Cuntz et al.[46]; importantly, our work shows that the parabolic scaling follows from stochastic branching (without making assumptions about optimal wiring that were made in the Cuntz model[46]).

Horizontal system (HS) cells in the central nervous systems of *Drosophila* and *Calliphora* show a similar scaling relationship[47]. Interestingly, the data from class IV dendrites fall on the same line as the

**Table 1 | Microscopic parameters associated with dynamic dendrite tips**

| Parameter | Symbol | Unit | 24 h AEL | 48 h AEL | 96 h AEL | Source |
|---|---|---|---|---|---|---|
| Branching rate | $k_b$ | min$^{-1}$·µm$^{-1}$ | 0.0082 ± 0.0017 | 0.0016 ± 0.0007 | 0.0009 ± 0.0006 | * |
| Growing speed | $v_G$ | µm·min$^{-1}$ | 1.61 ± 0.60 | 1.62 ± 0.67 | 1.64 ± 0.91 | * |
| Shrinking speed | $v_S$ | µm·min$^{-1}$ | 1.53 ± 0.59 | 1.08 ± 0.46 | 1.33 ± 0.61 | * |
| Transition rate | $k_{GP}$ | min$^{-1}$ | 0.784 ± 0.036 | 0.933 ± 0.044 | 0.923 ± 0.034 | * |
| Transition rate | $k_{GS}$ | min$^{-1}$ | 0.640 ± 0.033 | 0.435 ± 0.030 | 0.799 ± 0.032 | * |
| Transition rate | $k_{PG}$ | min$^{-1}$ | 0.335 ± 0.015 | 0.155 ± 0.007 | 0.116 ± 0.004 | * |
| Transition rate | $k_{PS}$ | min$^{-1}$ | 0.314 ± 0.014 | 0.235 ± 0.009 | 0.117 ± 0.004 | * |
| Transition rate | $k_{SG}$ | min$^{-1}$ | 0.598 ± 0.033 | 0.282 ± 0.022 | 0.575 ± 0.027 | * |
| Transition rate | $k_{SP}$ | min$^{-1}$ | 0.946 ± 0.041 | 1.251 ± 0.045 | 1.276 ± 0.040 | * |
| Drift velocity | $\bar{v}$ | µm·min$^{-1}$ | 0.038 ± 0.019 | 0.027 ± 0.018 | 0.022 ± 0.018 | ** |
| Diffusion | $D$ | µm$^2$·min$^{-1}$ | 0.5039 ± 0.0443 | 0.2673 ± 0.0691 | 0.0216 ± 0.0332 | ** |
| Rebranching probability | $\beta$ | – | 0.19 ± 0.04 | 0.17 ± 0.05 | 0.19 ± 0.02 | *** |
| Geometric factor | $\alpha$ | – | 1.564 | 1.462 | 1.359 | **** |
| Diffusion factor | $\gamma$ | - | 0.602 | 0.580 | 0.678 | **** |

The subscripts G, S, and P denote growing, shrinking, and paused states. The transition rate $k_{GP}$ denotes the rate of transition from the growing to the paused states. The errors are SEs except for the branching rates and speeds, and rebranching probability which are SDs.
* Shree et al.[20]. ** Calculated from the transition matrix (Methods). *** Fig. 3d. **** See text and Methods.

data from the horizontal-cell dendrites (Fig. 5d), which are also quasi-two-dimensional. This raises the possibility that horizontal cells develop by a similar tip-driven process.

### Class IV cells have a small mesh size similar to regular polygonal tilings

To quantify the space-filling properties of planar meshes, we defined the mesh size as the diameter of a randomly placed circle that has a 50% chance of intersecting with a line in the mesh[48]. It is the median size of the "holes" in the mesh. A similar concept to mesh size, termed space coverage, is defined by Baltruschat et al.[30]. Class IV cells have a mean mesh size of 4 µm, which ranges from 2 µm to 7 µm over development (Fig. 5e, green circles). Because the mesh size is similar to the diameter of the tip of the ovipositor (5 µm at its base[41]), the geometry of the class IV cells is well suited for the avoidance reflex: an ovipositor barb has a > 50% chance of making a direct hit on a branch.

As a benchmark for the economy of meshes, we calculated the mesh size of regular tilings of the plane by triangles, squares, and hexagons (Fig. 5f, left column). Unexpectedly, all regular tilings have the same slope of $2 - \sqrt{2} \cong 0.59$ when the mesh size is plotted against the inverse of their length density (Methods: Tilings by regular polygons). Class IV cells (Fig. 5e, green circles), agent-based simulations of class IV cells[20] (Fig. 5e, blue squares), and Voronoi tessellations (Fig. 5e, purple diamonds), all fall on the same line with slope $2 - \sqrt{2}$. This means that for a given total length of branches, these different tilings have the same mesh size: they fill up the space equally economically. Equally spaced parallel lines and directed-rod simulations (Fig. 5e, orange circles) have a smaller slope of 0.5 and are, therefore, more economical, presumably because they do not contain branch points. Other meshes are less economical: randomly spaced parallel lines have a slope of $\ln 2 \cong 0.69$, and Delaunay triangulations have even larger slopes (Fig. 5e, red triangles).

These comparisons show that the class IV arbors economically cover the plane. Therefore, the bottom-up stochastic growth process, governed by purely local rules, generates global structures with optimal properties, similar to structures obtained using top-down design principles such as regular polygons and minimal-spanning-tree models[30].

### The model predicts the arbor expansion speed and the decay length of the moving front

To measure the arbor expansion rate, we fitted the arbor diameter in the LR and AP directions with cubic polynomial

regression and calculated the slope at 24, 48, and 96 h (Fig. 6a). The expansion speed, which is half of the slope, was larger in the LR direction than in the AP direction (Fig. 6e), consistent with the larger sizes of the cells in the LR direction (Fig. 1c–e). The speed decreased from 0.03-0.05 µm/min at 24 h and 48 h to 0.01-0.02 µm/min at 96 h. These expansion speeds are 30- to 100-fold slower than the "instantaneous" growing and shrinking speeds of the tips, showing that the timescale of arbor growth is well separated from the timescale of tip growth. To characterize the front shape, we fitted an exponential curve, $e^{-r/\lambda}$, to the peripheral length density (Fig. 6b) to obtain the decay length $\lambda$, which is ~4 µm throughout development (Fig. 6f).

To predict the front speed ($c$) and decay length ($\lambda$), we solved Eqs. (1)-(3) in a co-moving frame $z \equiv r - ct$. To facilitate solution, we made three well-justified assumptions in the distal front: we omitted the internal dendrite density (because terminal branches are in the majority), we dropped the collision term (because it is second order in density, which is small), we set the radial transport term $(\sin\theta/r)\partial/\partial\theta$ to zero (because it decays as $1/r$). Using an exponential trial solution, $e^{-z/\lambda}$, we obtained a constraint Eq. (16) on the allowed values of $c$ versus $\lambda$ as shown in Fig. 6c. The marginally stable solution[49] $c_m$ must satisfy $dc/d\lambda|_{c=c_m} = 0$. This condition pinpoints the emerging front speed $c_m$ as the minimal speed along the $c$-$\lambda$ curve, which corresponds to a singular solution for the decay length $\lambda_m$.

The model's predictions for the front speed and decay length are shown in Fig. 6e, f. To validate these theoretical predictions, we also solved the mean-field equations numerically (Fig. 6d, Methods: Numerical solution): the theory and numerical solutions were in good agreement (Fig. 6e, f). The predicted front speed aligns well with the experimental values at 48 h. However, the predictions exceed the measurements by approximately two-fold at 24 and 96 h. We believe that the slower-than-predicted front speed at 24 h is partly due to asymmetric growth: on the LR axis, growth preferentially takes place towards the center line, reducing the expansion speed by roughly half (Supplementary Fig. 5a). The slower-than-predicted front speed at 96 h can be explained by dendrite expansion being limited by the diameter of the segment, which grows more slowly than the arbor can expand (Supplementary Fig. 5b). The predicted decay lengths are in good agreement with the experiments at all developmental ages (Fig. 6f). In

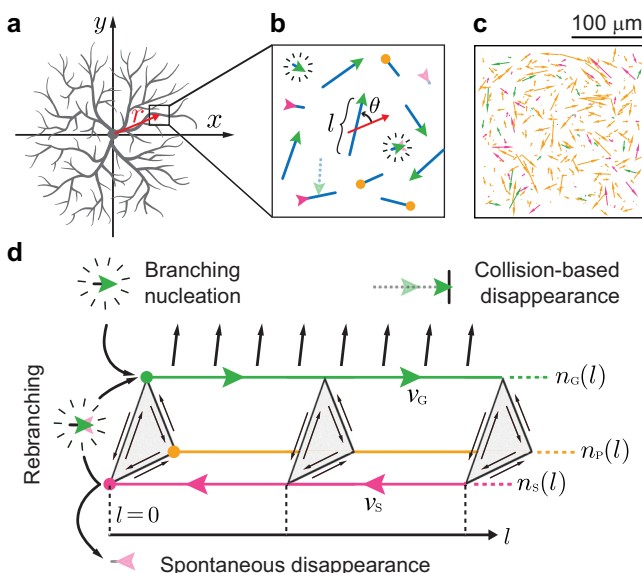

**Fig. 4 | Mean-field model. a** Schematic of a radially symmetrical class IV dendrite. The red arrow shows the radial direction within the black-box region. **b** Zoomed-in view of tip dynamics. Terminal branches (blue) transition between growing (green), shrinking (magenta), and paused (orange) states. Branches are lost in two ways. Either they collide with their base (length zero) in the shrinking state, or they collide with another branch and disappear. New branches are born by spontaneous branching or by regrowing following shrinkage to zero length ("rebranching", green arrowhead with radiating lines on top of a faded magenta arrowhead). $l$ and $\theta$ denote branch length and radial angle, respectively. **c** An example frame from the directed-rod simulation. **d** Schematic of the mean-field model. Dendrite densities of the growing (green), shrinking (magenta), and paused (orange) states are represented along the branch length axis with their corresponding colors. Dependence of dendrite densities on radial angle, radial position, and time are omitted. Green arrowheads represent the flow of growing dendrite density toward longer branch lengths due to elongation. Magenta arrowheads represent the flow of shrinking dendrite density toward shorter branch lengths due to retraction. State transitions are shown by gray triangles perpendicular to the length axes. Collision-based disappearance of growing dendrites is represented by black parallel arrows pointing away from the growing length axis. Additionally, boundary fluxes from branching, rebranching, and spontaneous disappearance are plotted at the left zero-length boundary ($l = 0$).

summary, our model semi-quantitatively agrees with the expansion of class IV neurons, especially before tiling.

### Length fluctuations increase the expansion speed

To determine how tip stochasticity impacts expansion, we calculated how tip drift velocity ($\bar{v}$) and tip diffusion ($D$) (calculated from the transition rates, Methods: Drift and Diffusion) individually affect the expansion speed (Fig. 7a). Expansion can occur even when the drift velocity is less than zero (to the left of the vertical dashed line in Fig. 7a), showing that fluctuations can drive growth. We calculated the boundary in phase space that separates growth from no growth analytically (red line in Fig. 7a, Supplementary Fig. 6). The boundary curve is a 2D generalization of the critical concentration in the Dogterom-Leibler model of microtubule dynamics, which separates the regions of bounded and unbounded growth. Our analysis shows that most of the expansion speed of real neurons (black circle in Fig. 7a) can be attributed to the fluctuations: if tip diffusion is reduced to zero ($D = 0$), the expansion speed would decrease roughly four-fold from 0.054 μm/min to 0.014 μm/min (V in Fig. 7a,b), equivalent to the one-state model; whereas if the drift velocity is reduced to zero ($\bar{v} = 0$), the expansion speed would only decrease by 37% to 0.034 μm/min (II in Fig. 7a, b). The influence of tip dynamics on expansion speed is illustrated by the directed-rod simulations in Fig. 7b (Supplementary

**Table 2 | Predictions of the one-state model**

| Parameter | Formula |
|---|---|
| Steady-state dendrite number density ($N$) | $N = 2k_b/(\alpha\bar{v})$ |
| Steady-state dendrite length density ($\rho$) | $\rho = (\sqrt{2k_b/\bar{v}})/\alpha$ |
| Average branch length ($\bar{l}$) | $\bar{l} \equiv \rho/N = \sqrt{\bar{v}/2k_b}$ |
| Number-length parabola | $N = \alpha\rho^2$ |
| Expansion rate (c), decay length ($\lambda$) | $c = \bar{v}/2, \lambda = 0$ |

See Table 1 for definitions of parameters. $\alpha = 0.75$ is the collisional prefactor for the one-state model calculated from simulations (Methods). The factors of 2 in length and density reflect correction for internal branches.

Fig. 3b and Supplementary Fig. 7). In conclusion, fluctuations, which are rectified by branching, play a major role in driving the expansion of dendritic arbors.

### The model predicts radial branch orientation

The branches of class IV cells tend to be radially oriented[20] (Fig. 2f–h), especially internal branches. To understand this, we plotted the radial orientations in the frontal and central regions of the arbor at 48 h AEL (see data from other stages in Supplementary Fig. 8): within the frontal region (gray area in Fig. 8a), there is a clear radial orientation of both the terminal (blue) and internal (brown) branches (Fig. 8b). This is well-accounted for by the model (Fig. 8b, solid line): the radial orientation in the front arises because outwardly oriented branches grow fastest in the radial direction (due to the $\cos\theta$ in transport term) and are less likely to collide with other branches. In the central region, the model predicts that the radial orientation of terminal branches is lost due to isotropic branching, as observed (Fig. 8c, solid line). By contrast, internal branches retain radial orientation in the center (Fig. 8c, brown histogram); we propose that this is because the radial orientation of frontal terminal branches is "locked in" when they convert to more stable central internal branches. These conclusions are supported by the simulations (Fig. 8d–f).

### Discussion

Using a mean-field model, we have shown that the stochastic dynamics of dendrite tips generates many of the morphological properties of class IV da dendrites as they grow during larval development. These properties include the exponential distribution of branch lengths (Fig. 2d), the mean dendrite length (Fig. 5a), the dendrite length density (Fig. 5b), and the approximate parabolic scaling between dendrite number and length densities (Fig. 5c). It also predicts the tight spacing of the dendritic meshwork (Fig. 5e) and shows that they space-fill the segments as economically as regular tilings. The model accounts for the radial orientation of terminal branches in the proximal region and explains the radial orientation of internal branches in the distal region if we make the additional, reasonable assumption that the orientation of internal branches is locked in by the branching process. Additionally, the model also shows that stochastic dendrite dynamics accounts for the expansion of the arbor and that fluctuations, caused by transition between growing, shrinking, and paused states, increase the expansion rate. Importantly, there are no free parameters: the microscopic parameters suffice to specify the macroscopic dynamic morphology.

The mean-field model provides analytic expressions of the geometric properties in terms of the microscopic parameters. While the analytic expressions are complicated for the full three-state model, the one-state model gives simple formulae: the average length is related to the branching rate ($k_b$) and the tip growth rate ($\bar{v}$)

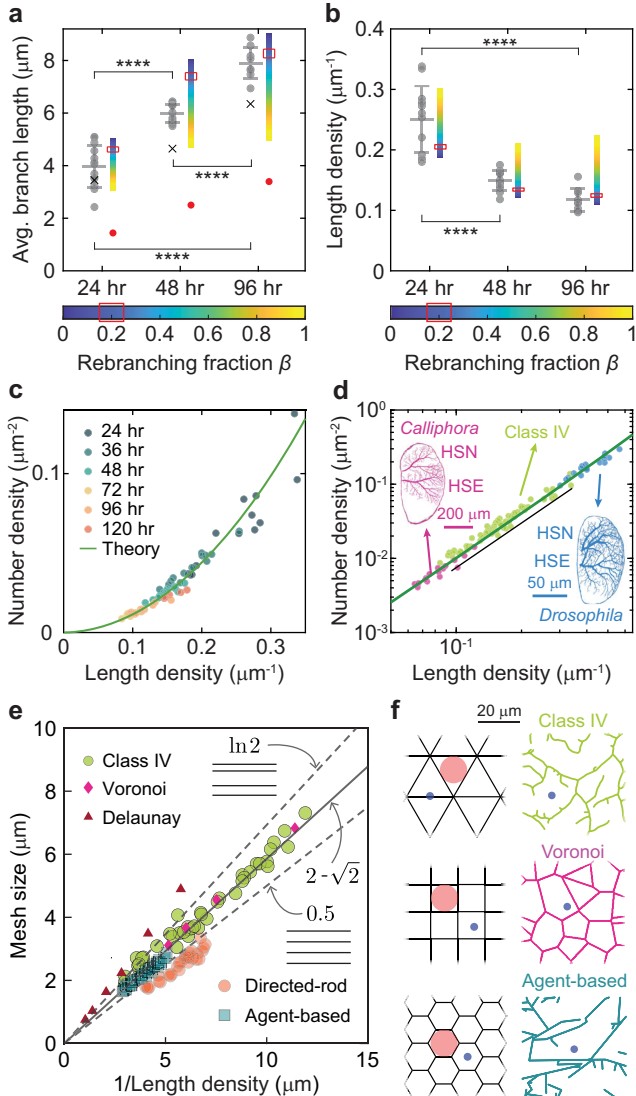

**Fig. 5 | The mean-field model predicts the steady-state branch lengths and densities. a** Experimental measurements (gray circles) of average branch length at 24 h (13 cells), 48 h (12 cells), and 96 h (9 cells) AEL. The means and standard deviations are superimposed as vertical bars. The model predictions with rebranching probabilities, β, varying from 0 to 1 are visualized by color-coded bars; the measured value of 0.2 is indicated by the red rectangle. The crosses are the characteristic lengths of the measured exponential distributions. Predictions from the one-state model are shown in red circles. See Methods for statistical analysis. **b** Measurements and model predictions of average length density. See (**a**) for details. **c** The parabolic relation between number density and length density. Each point represents a different cell. The solid line is predicted by the model using the parameters at 48 h AEL in Table 1. Parameters from 24 and 96 h AEL produce similar results. **d** The same data are plotted on a log-log axis, together with data from Horizontal System (HS) cells from *Drosophila melanogaster* and from the larger fly, *Calliphora erythrocephala*[47,75]. The model prediction is shown as the green line. The linear fit to the class IV data is $N_{tot} = \alpha \rho_{tot}^p$ with $\alpha = 0.84$ (± 2SE range 0.69 to 0.98) and $p = 1.84$ (± 2SE range 1.75 to 1.94). The fit to the directed-rod simulations (shown as the black line) has $\alpha = 1.03$ (± 2SE range 1.02 to 1.04) and $p = 2.01$ (± 2 SE range 2.01 to 2.02). **e** Mesh sizes for several tilings listed in the legend. The black solid line corresponds to regular 2D tilings, while the lower and upper dashed lines represent equally spaced and randomly spaced horizontal line patterns, respectively. **f** Examples of tilings of the plane by regular and irregular shapes. All tilings have the same length density. The red circles indicate the inscribed circles of the polygonal tilings. The blue circles represent the characteristic mesh size, defined as circles with a 50% chance of intersecting the branches.

by $\bar{l} = \sqrt{\bar{v}/2k_b}$, the branch number density is proportional to $k_b/\bar{v}$ and the arbor expansion rate is $\bar{v}/2$. Such simple expressions illustrate the benefit of the mean-field approach over the agent-based approach.

There are two other important conclusions. First, because the model contains no interactions with the substrate or other cells, the morphogenesis of class IV cells can be accurately modeled by a self-autonomous process. The one exception is that signals arising from the adjacent endothelium restrict the cells to their segments[50], so that growth is slowed after 48–72 h, when tiling is complete. Extracellular cues have been shown to be important for dendrite morphogenesis in other cells, such as Purkinje cells[51]. We expect these external signals will add complexity to the stochastic dendrite growth model by making the dynamics dependent on location within the arbor. Second, because the mean-field model has no topology (the arbor is a set of disconnected rods), hierarchical branching mechanisms, as postulated by Baltruschat et al.[30], are not necessary in the determination of class IV dendrite geometry. This is an important feature of the mean-field approach that distinguishes it from agent-based models[20], which are inherently topological. In summary, we have shown that stochastic tip dynamics is an economical and rapid space-filling mechanism for building dendritic arbors, without external guidance or hierarchical branching mechanisms. Our model provides a general theoretical framework for understanding how macroscopic branching patterns emerge from microscopic dynamics.

Our findings are expected to generalize to other arbors because the branching, elongation, and retraction mechanism (including contact-based retraction) has been observed in many other neuronal types (see references in the Introduction), though the specific parameters of dendrite growth will lead to cell-type dependent morphological variations (https://neuromorpho.org[52]). However, there are important differences between class IV arbors and the arbors of other cells that restrict the general applicability of our model. For example, hierarchical branching is certainly important in other cells. *Drosophila* Class III da neurons have actin-rich branchlets along the backbone of their dendrite[29,44,53] and grow by a back-bone first mechanism[29]. The branchlets may be analogous to the dendritic spines of vertebrate neurons such as Purkinje and pyramidal cells. The mean-field model could be generalized by adding an additional class of branches that do not themselves branch and that spontaneously transition (catastrophe) to a long-lived shrinking state. The number would depend on the branching rate, and the average length would equal to the growth rate divided by the catastrophe rate. Embryonic *Drosophila* class I cells[22,23] and *C. elegans* PVD sensory cells[31] have a clear hierarchy of primary, secondary, and tertiary branches.

The present model and the agent-based model of Shree et al.[20] are inherently two-dimensional; direct cell-cell collisions take place when the growing dendrite and the target dendrite grow on a 2D surface, as is the case for class IV dendrites[39]. Collision is crucial because it provides negative feedback that keeps the dendrite density finite. In 3D, however, direct collisions will be much rarer because a growing line has a zero probability of colliding with a fixed line. While dendrites have non-zero thickness, the collision rates will still be very small, and a contact-based retract mechanism alone would lead to densities much higher than observed. Therefore, if a collision-based retraction mechanism were to operate in 3D, the effective size of the growing tip would need to be increased. This could be done by using filopodia, actin-based structures, to reach out from the growing tip and detect nearby dendrite branches. Filopodia are well known in the growth cones of axons, and also found in vertebrate dendrites[12,15] and adult fly dendrites[54], though not in larval

class IV dendrites. Morphogens that can diffuse through the tissue could also provide signaling cues that lead to retraction in 3D. Thus, generalized contact-based retraction mechanisms might restrict dendrite density during the development of 3D arbors.

Our model provides molecular insights into dendrite morphology. Given the importance of branching for function, there has been a large effort to identify the molecular mechanisms underlying the formation, growth, and stability of dendrites during development[4,19,55–59]. Altered dendrite morphology often occurs when the expression of cytoskeletal, membrane-trafficking, and cell-adhesion molecules are perturbed. However, the mechanisms by which these perturbations alter dendritic geometry (e.g., branch length and density, arbor size) are not known. Our work establishes a morphogenetic "pathway" that causally links tip properties to arbor geometry. This reduces the genotype-to-phenotype problem to finding the molecular mechanisms that determine the dynamical properties of the dendrite tip. The model can then be used to extrapolate molecular perturbations of tip dynamics to those of the whole arbor, as demonstrated above using the one-state model to relate tip branching and growth to average branch length and density. Thus, the model generates etiological hypotheses.

We give a few examples. The *Drosophila* microtubule-associated protein minispindles (msps) is member of the chTOG/EXMAP215 family of proteins that accelerate microtubule growth[60]. If microtubules play a role in dendrite growth, we might expect that knocking down msps would slow tip growth, leading to shorter dendrites, a lower dendrite number density, and slower dendrite expansion (Table 2). By contrast, the kinesin-family proteins kinesin-8 and kinesin-13 are microtubule depolymerases that increase the catastrophe rate of growing microtubules[61]. Again, if microtubule dynamics drives dendrite dynamics, then we expect that knocking down the *Drosophila* members of these kinesin subfamilies will decrease the transition of growing microtubules to shrinking ones, leading to the opposite phenotypes to msps knockdown. Perturbations of actin growth[62], endocytosis[63], and lipid metabolism[64] perturb morphology, and the models discussed here make hypotheses for how they might affect the tip parameters, which can be tested experimentally. Thus, a morphogenetic "pathway" can be used to interpret molecular phenotypes.

Dendrite growth shares features with branching morphogenesis in other biological systems. A clear analogy is with cytoskeletal networks. Our model generalizes the 2-state Dogterom-Leibler model of microtubule dynamic instability[45] by adding extra states, branch formation (which is observed for both actin filaments[65] and microtubules[66]), and collision-based retraction. Our model generates radial expansion of dendritic networks similar to the traveling-wave-like expansion of microtubule arrays[67,68]. Therefore, our model is likely to be a useful generalization for describing cytoskeletal arrays.

Our model may also prove useful for analyzing the branching of tissues, which is also driven by dynamic structures. While branching is highly stereotyped in the lungs[69] and kidneys[70] and less stereotyped in insect trachea[71], branching in other tissues such as mammary glands, is highly variable, reminiscent of neurons[72]. In mammary glands, the leading front of branch tips has been described as a "branching engine… that initiates, directs, and maintains branch outgrowth… during development and regrowth"[73]. This description aptly describes dendrite tips as well.

## Methods

### Fly Stocks and maintenance

The fly line;;ppk-cd4-tdGFP (homozygous) was used to image class IV dendritic arborization neurons and was generously provided by Dr. Chun Han (Cornell University). Fly crosses were maintained in Darwin chambers set at 25 °C, 60% humidity with a 12 h light/dark cycle.

Embryos were collected on apple-agar plates, with a large drop of yeast paste placed at the center to stimulate egg-laying.

### Sample preparation

For larval imaging, larvae at 24, 48, 72, 96, and 120 h AEL were washed with 20% and 5% sucrose solutions, anesthetized using FlyNap (Carolina Biologicals, Burlington, NC, USA), and transferred to apple-agar plates for a 1–5 min recovery. Post-recovery, larvae were carefully positioned dorsal side up on a 1% agar bed affixed to a glass slide and imaged in a drop of 50% PBS and 50% halocarbon oil 700 (Sigma Aldrich). To immobilize larvae, a 22 mm × 22 mm coverslip lined with Vaseline or vacuum grease was gently pressed over them.

### Imaging

Samples were imaged using a spinning disk confocal microscope, specifically a Yokogawa CSU-W1 disk (pinhole size 50 μm) integrated into a fully automated Nikon TI inverted microscope with perfect focus. Excitation was achieved using a 488 nm laser at 18–21% power, and imaging was performed with either a 40X (1.25 NA, 0.1615-micron pixel size) or 60X (1.20 NA, 0.106-micron pixel size) water immersion objective. Images were captured with an sCMOS camera (Zyla 4.2 Plus) and processed using Nikon Elements software. Prior to imaging, samples were manually focused to identify abdominal third and fourth segment (A3 or A4) neurons.

For density analysis, static images were acquired using a 60X water immersion objective for 24 h larvae and a 40X objective for later stages. Images were stitched using ImageJ. Image processing, including segmentation, skeletonization, and density measurements were conducted using in-house MATLAB algorithms whereas` branching/rebranching analysis was done manually using ImageJ[20]. For branching/rebranching and tip dynamics full-frame (2048 × 2048 pixels) movies with duration 15–30 min were acquired, containing 6–12 sections (~1 μm per section) collected at 4–6 s intervals.

### Rebranching probability (β)

We located the disappearance of a retracting branch (a debranching event) and defined a rebranching event as the appearance of another branch within ±2 pixels over the course of a 15–20 min recording. Rebranching usually occurred within 5 min (Supplementary Fig. 1e), which is much shorter than the time expected if the new branched were formed spontaneously, given the branching rate in Table 1. The rebranching probability, β, is the number of rebranching events divided by the number of debranching events.

### Densities in the central region

We calculated the dendrite number and length densities at different developmental stages from skeletonized images. First, the neurons are aligned in the AP and LR direction and then binarized using an in-house MATLAB algorithm. Second, the binarized image was skeletonized using the MATLAB function 'bwmorph'. Assuming, the mass of the dendrite skeleton is uniformly distributed in a rectangle, we determined the neuron widths in AP and LR directions as $D_{AP} = \sqrt{12}\, R_g^{AP}$ and $D_{LR} = \sqrt{12}\, R_g^{LR}$, where $R_g = \left(\frac{1}{N}\sum_{j=1}^{N}\left(r_j - \bar{r}\right)^2\right)^{1/2}$ is the radius of gyration. $N$ is the total number of occupied pixels in the skeleton, $r_j$ is the projection onto the respective axis of the $j^{th}$ occupied pixel and $\bar{r}$ is the position of the center of mass. To calculate the densities in the central region, we removed the peripheral 5% of the neurons from all four sides. We calculated the total branch length and number of tips in this trimmed region and normalized these values by the containing area to calculate the branch length and tip densities[20]. To calculate the densities in AP and LR directions (as shown in Fig. 1f,g) the total skeletal mass was projected on these axes and subsequently normalized by the pixel length.

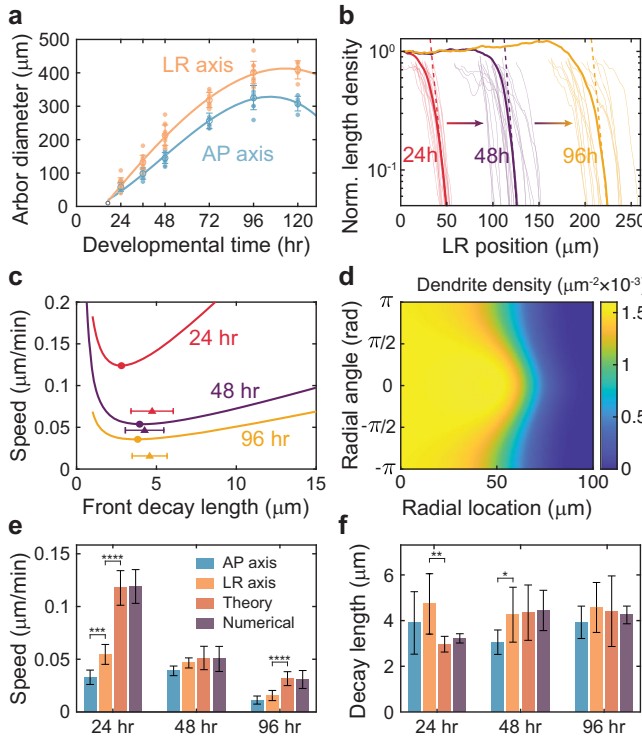

**Fig. 6 | The mean-field model reveals that arbor expansion is driven by length fluctuations. a** Measured arbor diameters are plotted from 24–120 h AEL along the anterior-posterior (AP) and left-right (LR) axes. Growth initiates at 16 h AEL from a soma of size 10 μm (black hollow circle). Cubic regressions used to fit the front speed are shown in solid lines. **b** Dendrite length density along the LR axis is normalized by its central density. Solid lines show the mean front profiles at 24 (red), 48 (purple), and 96 (yellow) hours AEL, averaged from individual neuron front profiles shown in faded lines aligned at 10% of their central length density. The lower 25% of the densities are fitted with exponential functions (dashed lines) to estimate the decay length. **c** Allowed values of the front speed $c$ and decay length $\lambda$ by Eq. (16). The marginally stable solution coincides with the global minimum of the front speed confined on the $c$-$\lambda$ curve (solid circles). The average measured front speed and decay length along the LR axis are visualized by solid triangles. **d** Numerical solutions of the dendrite density at 1600 min from an initial isotropic sigmoid front of radius -10 μm. The computation uses parameters at 48 h AEL. **e**, **f** Front speed and front decay length from measurements along AP and LR axes, theoretical predictions, and numerical solutions at 24, 48, and 96 h AEL. All errors are standard deviations. Those for the theory are estimated by bootstrapping (Methods: Statistical Analysis).

## Mesh size

We defined the "mesh size" of a network of lines (branches) as the diameter of the circle that has a 50% chance of intersecting a branch[48]. To measure the mesh size, we randomly generated $N_{circle} = 5000$ circles with a diameter ($D$) and randomly placed them within the network. If a circle overlaps with any part of the network, we consider it as a "hit", with number $N_{hit}$. The hitting probability is $P_{hit}(D) = N_{hit}/N_{circle}$. By systematically calculating the hitting probability as a function of the diameter of hitting circles $D$, we determined the mesh size, $M$, such that $P_{hit}(M) = 0.5$. For very small diameters of the hitting circle, the hitting probability is close to zero. We extensively validated our method using the regular lattices: triangular, square, and hexagonal. As expected, the measured mesh size as a function of sparsity (inverse of length density) for these regular lattices fell on a line with a slope of $2-\sqrt{2} \cong 0.59$, which can be calculated from the geometry. We also measured the mesh sizes for equal-spaced horizontal lines (expected mesh/sparsity = 0.5), for exponentially-spaced horizontal lines (expected mesh/sparsity = ln2 $\cong$ 0.69), for Voronoi tessellations and Delaunay triangulation of random points, and minimum spanning

trees (MSTs) using the Trees Toolbox[6], varying the balancing factors (bf) from 0 to 1. Some of these results are plotted in Fig. 3e, along with the mesh sizes of class IV dendrites and simulated dendrites using the one- and three-state models.

## Front velocity (c) and front decay length (λ)

The arbor diameter was determined by measuring the radius of gyration for each neuron along the AP and LR axes at 24, 36, 48, 72, 96, and 120 h AEL. We assumed that growth started at 16 h after egg lay from a soma size of 10 μm. The expansion speed was estimated from the slope of the best-fit cubic curve to the diameter data divided by 2. To get an average front profile at 24, 48, and 96 h AEL, dendrite length densities from individual neurons were normalized to and aligned at 10% of their central plateau values. We measured the decay length by fitting an exponential curve to the average dendrite length density from the outermost edge to where the density falls to 25% of its steady state. Standard errors are obtained with bootstrapping.

## Radial branch orientation

The radial orientations of dendritic branches were measured at the branch midpoint. A straight line drawn from the soma to the branch midpoint defines the radial direction. Another line was fitted through a 1 μm segment near the branch midpoint to represent the tangential branch direction. The radial angle was measured from the radial direction to the branch direction, with positive angles conventionally defined in the counterclockwise direction.

## Statistical analysis

We performed ordinary one-way ANOVA tests with Tukey's correction for multiple comparisons (Figs. 3d, e and 5a, b) using GraphPad Prism (version 10), considering the number of animals as the sample size with 5% significance level. Only significant comparisons ($p < 0.05$) are reported in the figures. All data presented in Figs. 3d, e and 5a, b satisfied the normality test.

To calculate errors in front velocity measurements (Fig. 6a, e), we employed a bootstrapping method. We randomly sampled a measurement for arbor diameter at each time point, using which we calculated the front velocity from cubic regression. This process was repeated 1000 times to generate a distribution of front velocity estimates. The errors in the mean-field predictions and numerical solutions for the front velocity and front decay length were also obtained from bootstrapping (Fig. 6e, f). We used the measured mean values and standard errors of the dynamic parameters to generate 1000 sets of parameters. From these, we calculated the corresponding distributions of front velocities and decay lengths, allowing us to determine the uncertainty in our theoretical predictions and numerical solutions.

## Directed-rod simulations

**Simulation of the steady-state density.** A square box of size $L$ (200 μm) with periodic boundaries was initialized with $N_i$ rods whose bases $(x_i^b, y_i^b)$ were randomly and uniformly distributed within the box. The individual rods were assumed to be infinitely thin straight line-segments growing in random direction $\theta_i \in [0, 2\pi]$. The growing ends of the rods (tips) stochastically transition between growing (G), shrinking (S) and paused (P) states with experimentally measured rates ($k_{ij, j \neq i \in (G, S, P)}$) from Table 1. We divided the simulation time into small steps $\Delta t$ and implemented a standard 'Monte-Carlo' method such that the total probability of a transition in the time interval is $P_i = 1 - e^{-k_{tot}\Delta t}$, where $k_{tot}$ is the sum of the transition rates from one particular state: $k_{tot} = \sum_{j=(G,S,P)}^{j \neq i} k_{ij}$. Subsequently, $P_i$ is compared with a uniform random number $R(0, 1)$ to implement the transition. If there is a transition, it happens maintaining the ratio $k_{ij}/k_{tot}$. After the transition, the tip is assigned with corresponding mean state velocity ($V_G, V_S, V_P$). We calculate the length of the tips at time $t$ as follows:

$l_i(t) = l_i(t - \Delta t) + V_{(G,S,P)}\Delta t$, where $V_{(G,S,P)}$ is the mean velocity in the growing/paused/shrinking states. The end points of the rods are given by: $x_i^t(t) = x_i^b + l_i(t)\cos(\theta_i)$ and $y_i^t(t) = y_i^b + l_i(t)\sin(\theta_i)$ where, $x_i^t(t)$ and $y_i^t(t)$ represents the end-point location of the $i^{th}$ tip and $\theta_i$ is the random angle associated with the tip. In the simulations, we assumed that the rods disappear instantaneously after collision, consistent with our experimental study showing that after collision the branches shrink[20]. The lateral branching of the dendrite tip is conceptualized in our model as formation (birth) of a randomly oriented nascent branch at a random location within the simulation box (green arrowhead in Sup-

plementary Fig. 3a). We used experimentally measured branching rate $k_b$ ($\mu m^{-1}$ $min^{-1}$) to nucleate new tips in the system[20]. To implement this, we visit all the existing tips and calculate the branching probability $P_i^b = 1 - e^{-l_i(t)k_b\Delta t}$ for individual tips. We then, compare a uniform random number with the branching probability $P_i^b$ to nucleate a new tip at a random location within the box.

**Simulation of the expanding front.** The system is initialized by uniform nucleation of $N_{ini}$ number of tips within a circular region of diameter $D_{ini}$ (Supplementary Fig. 3b). Simulation of tip dynamics follows the same rules as in the case with periodic boundary conditions, except branching. New branches (green arrowhead) are nucleated at a random location on the circle passing through the midpoint of the mother branch.

**Fitting of collision parameters**
The dimensionless pre-factors $\alpha$ and $\gamma$ in the collision term were determined from the directed-rod simulations. $\alpha$ is a geometric pre-factor. $\gamma$ is a diffusion pre-factor that reflects the likelihood of tip collision before shrinking to its base The collision rate, defined by the total number of collisions per unit time normalized by the total number of branches, was calculated at several different branching rates $k_b$, ranging from 0.0005 to 0.01 $min^{-1}\cdot\mu m^{-1}$ (Supplementary Fig. 3c). The three-state collision rate depended quadratically on the length density and the fit to $\alpha\bar{v}\rho + \alpha^2\gamma D\rho^2$ was used to estimate $\alpha$ and $\gamma$ (Table 1). The one-state collision rate depended linearly on $\rho$ (Supplementary Fig. 3c), the $\alpha$ pre-factor value of which is fitted to be $\alpha_1 = 0.750$. The Péclet number, defined by the ratio of advective to diffusive collisions, is plotted for the three-state simulation is on the order of 1 (Supplementary Fig. 3d). The simulations recapitulate the parabolic relations between number density and length density (Supplementary Fig. 3e).

**Tilings by regular polygons**
For tilings by triangles, squares and hexagons of side $l$, the mesh sizes are $l(1-1/\sqrt{2})/\sqrt{3}$, $l(1-1/\sqrt{2})$, and $l\sqrt{3}(1-1/\sqrt{2})/2$ respectively.

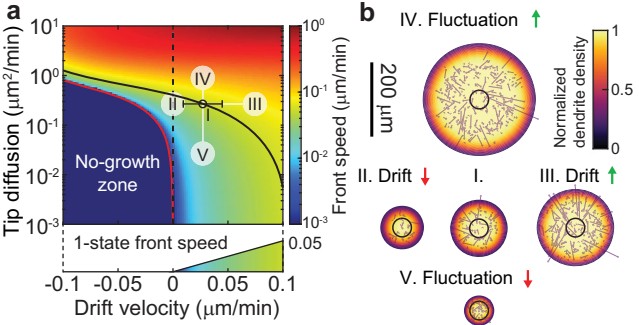

**Fig. 7 | Length fluctuation increases arbor expansion speed. a** Phase diagram of front speed as a function of the tip drift velocity (x-axis) and effective diffusion coefficient (y-axis). The black circle corresponds to the measured average tip speed at 48 h AEL. The error bar represents the standard error in tip speed at 48 h AEL. The black curve represents the contour of constant front speed passing through the black circle. The red curve indicates the phase boundary, which represents a threshold below which the arbor cannot grow. **b** Simulated arbor sizes after 1000 min using different mean tip velocities and length fluctuations indicated in (**a**). I: data at 48 h. II: zero average tip speed. III: increased average tip speed. IV: Increased fluctuations. V: Decreased fluctuations. The color code represents normalized dendrite density. Representative directed-rod simulations are superimposed. See growth dynamics of these simulations in Supplementary Fig. 7.

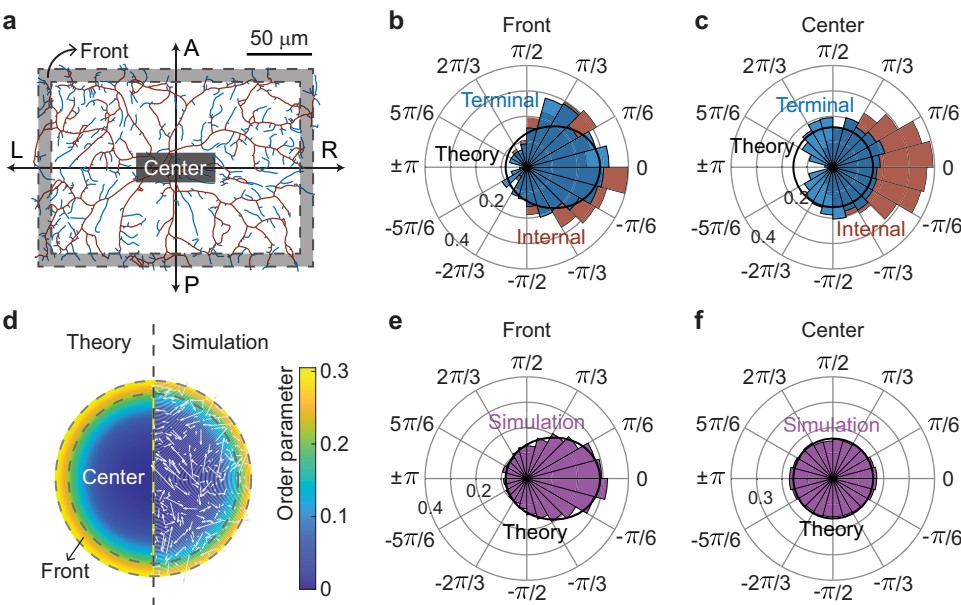

**Fig. 8 | Radial orientation of dendrite branches. a** The dendrites are separated into frontal and central regions. The same skeletonized neuron as in Fig. 2b is shown (48 h AEL). **b, c** Radial-angle distributions of terminal and internal branches at 48 h AEL within the frontal and central regions, respectively. The black curves represent mean-field predictions using 48-h parameters, agreeing with the terminal-branch distributions. **d** The radial orientation of dendrites computed

from directed-rod simulations is visualized by the order parameter $\left|\sum_{j=1}^{M} e^{i\theta_j}/M\right|$ (zero corresponds to a uniform radial distribution). The left and right halves depict, respectively, the order parameter from the mean-field theory and the directed-rod simulations. The right half is overlaid with a representative configuration snapshot from the simulation. **e, f** The directed-rod simulations (histograms) align closely with the mean-field theory (solid line).

The respective length densities are $(2\sqrt{3})/l$, $2/l$, and $(2/\sqrt{3})/l$. Remarkably, the ratios are $2 - \sqrt{2}$ for all three.

## Dendrite tip drift velocity ($\bar{v}$) and diffusion coefficient ($D$)

The tip drift velocity $\bar{v}$ and tip diffusion $D$ were calculated from the six transition rates $k_{ij}$ and two tip velocities $v_G$ and $v_S$ (Table 1). The steady-state probabilities of the tip being in the growing, shrinking, and paused state are: $P_G = (k_{PG}k_{SG} + k_{PG}k_{SP} + k_{PS}k_{SG})/P$, $P_S = (k_{GS}k_{PG} + k_{GP}k_{PS} + k_{GS}k_{PS})/P$ and $P_P = (k_{GP}k_{SG} + k_{GP}k_{SP} + k_{GS}k_{SP})/P$ where $P \equiv k_{PG}k_{SG} + k_{PG}k_{SP} + k_{PS}k_{SG} + k_{GS}k_{PG} + k_{GP}k_{PS} + k_{GS}k_{PS} + k_{GP}k_{SG} + k_{GP}k_{SP} + k_{GS}k_{SP}$ is a normalization factor. The drift velocity is: $\bar{v} \equiv P_G v_G - P_S v_S$.

The diffusion coefficient $D$ was calculated using the Green-Kubo relation, which simplifies to $D = \int_0^\infty \left[ \langle v(t+t')v(t') \rangle_{t'} - \bar{v}^2 \right] dt$ at the steady state[74]. The first term in the integral $\langle v(t+t')v(t') \rangle_{t'}$ is the autocorrelation function of the tip velocity. Making use of the Markov property and ergodic hypothesis, it can be shown that $\langle v(t+t')v(t') \rangle_{t'} = \sum_{i,j \in (G,S,P)} v_i v_j P(i,t|j,0)P_j$ where $P(i,t|j,0)$ is the conditional probability that the tip is in state $i$ at time $t$ conditioned on it was initially in state $j$ at time 0. The expression for $P(i,t|j,0)$ can be found explicitly by solving the linear equations of state transition:

$$\frac{d\mathbf{P}}{dt} = \mathbf{K}_{TR}\mathbf{P}, \quad \mathbf{K}_{TR} \equiv \begin{pmatrix} -(k_{GS}+k_{GP}) & k_{SG} & k_{PG} \\ k_{GS} & -(k_{SG}+k_{SP}) & k_{PS} \\ k_{GP} & k_{SP} & -(k_{PG}+k_{PS}) \end{pmatrix}. \tag{7}$$

The transition rate matrix $\mathbf{K}_{TR}$ has three eigenvalues, one is zero, whose eigenvector can be the steady-state probability $\mathbf{w}_0^T = \mathbf{P}_{SS}^T \equiv (P_G, P_S, P_P)$. The other two eigenvalues are both negative, denoted by $-\lambda_1$ and $-\lambda_2$, with their corresponding eigenvectors as $\mathbf{w}_1 \mathbf{w}1$ and $\mathbf{w}_2$. By decomposing an initial probability in state $j$ $\mathbf{P}_0^{(j)}$ P0j as $\mathbf{P}_0^{(j)} = \mathbf{P}_{SS} + a_1^{(j)}\mathbf{w}_1 + a_2^{(j)}\mathbf{w}_2$, the distribution probability at time $t$ becomes $\mathbf{P}(t) = \mathbf{P}_{SS} + a_1^{(j)}\mathbf{w}_1 e^{-\lambda_1 t} + a_2^{(j)}\mathbf{w}_2 e^{-\lambda_2 t}$. Hence the conditional probability can be written as $P(i,t,|j,0) = P_i + a_1^{(j)}w_{1,i}e^{-\lambda_1 t} + a_2^{(j)}w_{2,i}e^{-\lambda_2 t}$. As a result, the autocorrelation function becomes $\langle v(t+t')v(t') \rangle_{t'} = \bar{v}^2 + \sum_{i,j \in (G,S,P)} v_i v_j P_j \left( a_1^{(j)}w_{1,i}e^{-\lambda_1 t} + a_2^{(j)}w_{2,i}e^{-\lambda_2 t} \right)$. After taking the integral in the Green-Kubo relation, the diffusion coefficient satisfies $D = \sum_{i,j \in (G,S,P)} v_i v_j P_j \left( a_1^{(j)}w_{1,i}/\lambda_1 + a_2^{(j)}w_{2,i}/\lambda_2 \right)$. These calculations were checked by simulations.

## Steady-state solution of the three-state model

To determine the homogeneous steady-state solution, we begin by setting all time derivatives and spatial transport terms to zero in Eqs. (1-3). By expressing $n_P$ in terms of $n_G$ and $n_S$ using Eq. (3), we reduce the system to a two-dimensional matrix equation for $n_G$ and $n_S$:

$$\frac{d}{dl}\begin{pmatrix} n_G \\ n_S \end{pmatrix} = \begin{pmatrix} -\frac{K_{col}\chi_{GP}}{v_G} - \frac{K_{GS}}{v_G} & -\frac{K_{col}\chi_{SP}}{v_G} + \frac{K_{SG}}{v_G} \\ -\frac{K_{GS}}{v_S} & \frac{K_{SG}}{v_S} \end{pmatrix}\begin{pmatrix} n_G \\ n_S \end{pmatrix} \tag{8}$$

Here $\chi_{GP} = 1 + k_{GP}/(k_{PG}+k_{PS})$, $\chi_{SP} = 1 + k_{SP}/(k_{PG}+k_{PS})$, $K_{GS} = k_{GS} + k_{GP}k_{PS}/(k_{PG}+k_{PS})$, and $K_{SG} = k_{SG} + k_{SP}k_{PG}/(k_{PG}+k_{PS})$. The coefficient matrix, denoted by $\mathbf{K}$, has two real eigenvalues of opposite signs, as indicated by its negative determinant. Since $n_G$ and $n_S$ must asymptotically approach zero as $l \to \infty$ to be physically meaningful, the positive eigenvalue should be discarded. Denoting the negative eigenvalue as $-|\nu|$, the solution takes the form as $n_G(l) = n_G(0)\exp(-|\nu|l)$ and $n_S(l) = n_S(0)\exp(-|\nu|l)$. These solutions reveal that the growing and shrinking branches follow exponential length distributions with a characteristic mean length of $\bar{l} = 1/|\nu|$. The

vector $[n_G(0), n_S(0)]^T$ is an eigenvector corresponding to the eigenvalue $-|\nu|$. By combining the eigenequation with the boundary condition Eq. (6), we can derive explicit expressions for $n_G(0)$ and $n_S(0)$ in terms of $\rho_{tot}$ and $\bar{l}$:

$$n_G(0) = \frac{K_{SG} + v_S/\bar{l}}{K_{SG}v_G - \beta K_{GS}v_S + v_G v_S/\bar{l}}k_b\rho_{tot}, \tag{9}$$

$$n_S(0) = \frac{K_{GS}}{K_{SG}v_G - \beta K_{GS}v_S + v_G v_S/\bar{l}}k_b\rho_{tot}. \tag{10}$$

Substituting the expressions for $n_G(0)$ and $n_S(0)$ derived above into the definition of total length density $\rho_{tot} = 2\int_0^\infty dl\, l[n_G(l) + n_S(l) + n_P(l)]$ and canceling $\rho_{tot}$ from both sides, we obtain a self-consistent cubic equation for $\bar{l}$:

$$\frac{1}{\bar{l}^3} + \frac{A}{\bar{l}^2} = \frac{B}{\bar{l}} + C, \tag{11}$$

where $A$, $B$, and $C$ are given by $A = K_{SG}/v_S - \beta K_{GS}/v_G$, $B = 2k_b\chi_{GP}/v_G$, and $C = 2k_b(\chi_{GP}K_{SG} + \chi_{SP}K_{GS})/v_G v_S$. The cubic equation has a single positive root, corresponding to our desired value of $\bar{l}$. The dendrite length density $\rho$ can be further obtained from the following quadratic equation derived from $\det(\mathbf{K} + \mathbf{I}/\bar{l}) = 0$ (where $\mathbf{I}$ is the identity matrix):

$$\alpha\bar{v}\rho_{tot} + \gamma D\alpha^2\rho_{tot}^2 = \frac{v_G}{\chi_{GP}\bar{l}} \frac{1 + (K_{SG}/v_S - K_{GS}/v_G)\bar{l}}{+ (K_{SG}/v_S + \chi_{SP}K_{GS}/\chi_{GP}v_S)\bar{l}^2}. \tag{12}$$

The total dendrite number density is given by $N_{tot} = \rho_{tot}/\bar{l}$.

## Steady-state solution of the one-state model

In the one-state model, branches grow with a constant drift velocity $\bar{v}$ without fluctuation and disappear only by collision. This corresponds to the limit where the state transitions occur at high frequency. The mean-field equation reduces to $\partial_t n(r,l,\theta,t) = -\alpha\bar{v}\rho_{tot}(r,t)n(r,l,\theta,t) - \bar{v}\partial_l n(r,l,\theta,t) - \bar{v}\mathcal{R}(r,\theta)n(r,l,\theta,t)/2$ where the subscripts refer to partial differentiation. In the homogeneous isotropic steady state, the dendrite density only depends on branch length $l$. Thus, we can denote dendrite density as $n(l)$ by keeping only the length variable. Consequently, the one-state equation reduces to $\partial_l n(l) = -\alpha\rho_{tot}n(l)$. Here, $\rho_{tot}$ can be regarded as an unknown constant (independent of branch length $l$) that the total length density reaches at steady state. The solution is therefore $n(l) = n(0)e^{-\alpha\rho_{tot}l}$. Following the definitions for the total number and length density by taking the zeroth and first moments, we obtain two relations: $N_{tot} = 2n(0)/(\alpha\rho_{tot})$ and $\rho_{tot} = 2n(0)/(\alpha\rho_{tot})^2$. This gives the parabolic relation $N_{tot} = \alpha\rho_{tot}^2$. The average branch length is $\bar{l} = 1/(\alpha\rho_{tot})$. In the absence of shrinking branches, the boundary condition at length zero ($l = 0$) requires a balance between new tip generation and transport by drift velocity: $k_b\rho_{tot}(t) = \bar{v}n(l = 0, t)$. Substituting $n(0) = \alpha^2\rho_{tot}^3/2$ (re-writing $\rho_{tot} = 2n(0)/(\alpha\rho_{tot})^2$) into the boundary condition, we find that $\rho_{tot} = \sqrt{(2k_b/\bar{v})}/\alpha$ and $\bar{l} = \sqrt{\bar{v}/2k_b}$. These relations are summarized in Table 2.

## Relaxation time to the steady state of the three-state model

By introducing sudden changes in the parameter sets at 48 and 96 h AEL, we observed relaxation times of 1–4 h from numerical solutions of Eqs. (1–3) (Supplementary Fig. 4a–c). This timescale is much shorter than the durations of the three larval stages, which are 1 or 2 days. We also varied the microscopic parameters linearly from 24−48 h and 48−96 h. As expected, the numerical solutions closely followed the steady-state curve calculated by the parameter set at each time point (Supplementary Fig. 4d−f). This suggests that the dendritic branch density in the center of the arbor can be considered to remain in steady

state throughout larval development. Furthermore, we found that the one-state model equilibrates on a timescale similar to the three-state model (Supplementary Fig. 4g–i), indicating that the relaxation behavior is primarily affected by branching and collision rates, rather than tip transition dynamics, which equilibrate within minutes.

## Time-dependent solution of the one-state model

To solve the time-dependent solution, we reformulate the one-state equation to the governing equations of the branch number density $dN_T/dt = k_b\rho_T - \alpha\bar{v}\rho_T N_T$ and length density $d\rho_T/dt = \bar{v}N_T - \alpha\bar{v}\rho_T^2$. These equations can be derived by taking the $0^{th}$ and $1^{st}$ order moments of $l$ on both sides of the one-state equation $\partial_t n(l,t) = -\alpha\bar{v}\rho_T n - \bar{v}\partial_l n$. Given the average branch length $\bar{l} \equiv \rho_T/N_T$, quite surprisingly, we found that $\bar{l}$ satisfies an autonomous differential equation.

$$\frac{d\bar{l}}{dt} = \frac{d\rho_T}{dt}\frac{1}{N_T} - \frac{\rho_T}{N_T^2}\frac{dN_T}{dt} = \bar{v} - k_b\bar{l}^2 \tag{13}$$

which is independent of the geometric pre-factor $\alpha$. Solving the equation analytically yields that:

$$\frac{|\bar{l}(t) - \bar{l}_{SS}|}{\bar{l}(t) + \bar{l}_{SS}} = e^{-2\sqrt{k_b\bar{v}}t}\frac{|\bar{l}(t=0) - \bar{l}_{SS}|}{\bar{l}(t=0) + \bar{l}_{SS}} \tag{14}$$

Here SS denotes steady state. The characteristic time scale is $\tau = 1/2\sqrt{k_b\bar{v}} = 1/(2k_b\bar{l}_{SS})$, which is about 28, 76 and 112 min for 24, 48 and 96-h AEL parameter set. In retrospect, $1/\sqrt{k_b\bar{v}}$ produces the only timescale from the viewpoint of dimensional analysis.

## Parabolic relation between dendrite number density and length density

In the limit of small length density $\rho_{tot}$ and long average branch length $\bar{l}$, the right-hand side of Eq. (12) simplifies to $(K_{SG}v_G - K_{GS}v_S)/\bar{l}(\chi_{GP}K_{SG} + \chi_{SP}K_{GS}) = (P_G v_G - P_S v_S)/\bar{l} = \bar{v}/\bar{l}$. If the second-order diffusive collision term on the left-hand side of Eq. (12) is left out, one arrives at $\alpha\rho_{tot} = 1/\bar{l}$ with $\bar{v}$ canceled out from both sides. Coupled with $\bar{l} = \rho_{tot}/N_{tot}$, it becomes evident that $N_{tot} = \alpha\rho_{tot}^2$, which is identical to the parabolic relation between dendrite number density and length density derived from the one-state model.

## Traveling wave solution for the expanding arbor

With an exponential ansatz $e^{-z/\lambda}$ in the co-moving frame $z = r - ct$, one can rewrite Eqs. (1–3) as follows:

$$\frac{\partial}{\partial l}\begin{pmatrix} n_G \\ n_S \end{pmatrix} = \begin{pmatrix} \frac{\cos\theta}{2\lambda} - \frac{1}{v_G}\left(\frac{c}{\lambda}\chi_{GP}(c,\lambda) + K_{GS}(c,\lambda)\right) & \frac{K_{SG}(c,\lambda)}{v_G} \\ -\frac{K_{GS}(c,\lambda)}{v_S} & \frac{\cos\theta}{2\lambda} + \frac{1}{v_S}\left(\frac{c}{\lambda}\chi_{SP}(c,\lambda) + K_{SG}(c,\lambda)\right) \end{pmatrix}\begin{pmatrix} n_G \\ n_S \end{pmatrix} \tag{15}$$

Here, we modified the notations of $\chi_{GP}$, $\chi_{SP}$, $K_{GS}$, and $K_{SG}$ to incorporate the dependence on front velocity $c$ and decay length $\lambda$: $\chi_{GP}(c,\lambda) = 1 + k_{GP}/(c/\lambda + k_{PG} + k_{PS})$, $\chi_{SP}(c,\lambda) = 1 + k_{SP}/(c/\lambda + k_{PG} + k_{PS})$, $K_{GS}(c,\lambda) = k_{GS} + k_{GP}k_{PS}/(c/\lambda + k_{PG} + k_{PS})$, and $K_{SG}(c,\lambda) = k_{SG} + k_{SP}k_{PG}/(c/\lambda + k_{PG} + k_{PS})$. Notice that we have left out the internal branch density near the dendrite periphery, the collision term $K_{col}(r,t)n_G$ as a higher-order density term, and the radial transport term $(\sin\theta/r)\partial\theta$ due to its $1/r$ decay. The eigenvalues $q_\pm$ of Eq. (15) are $q_\pm = \cos\theta/2\lambda + P(c,\lambda) \pm \sqrt{P^2(c,\lambda) + Q(c,\lambda)}$, where we define $P(c,\lambda) \equiv [\chi_{SP}(c,\lambda)c/\lambda + K_{SG}(c,\lambda)]/2v_S - [\chi_{GP}(c,\lambda)c/\lambda + K_{GS}(c,\lambda)]/2v_G$ and $Q(c,\lambda) \equiv [\chi_{SP}(c,\lambda)c/\lambda + K_{SG}(c,\lambda)][\chi_{GP}(c,\lambda)c/\lambda + K_{GS}(c,\lambda)]/v_G v_S - K_{SG}(c,\lambda)K_{GS}(c,\lambda)/v_G v_S$. Considering $P + \sqrt{P^2 + Q} > 0$ since $Q$ is positive, the

larger eigenvalue $q_+$ must be discarded. Otherwise, $q_+$ can be positive for some values of $\theta$, resulting in unrealistic divergence of branch density as branch length $l$ approaches infinity. With $q_-$ being the only permissible eigenvalue, the branch densities $n_G$ and $n_S$ can be expressed by $n_G(z,l) = n_G(0)e^{q_- l - z/\lambda}$ and $n_S(z,l) = n_S(0)e^{q_- l - z/\lambda}$. Integrating over $l$ and $\theta$ as per Eq. (5), we could express length density $\rho$ in terms of $n_G(0)$, $n_S(0)$, $\chi_{GP}(c,\lambda)$, $\chi_{SP}(c,\lambda)$, $P(c,\lambda)$, and $Q(c,\lambda)$. One can eliminate $\rho$ with the boundary condition in Eq. (6) and obtain a fractional constraint on $n_G(0)$ and $n_S(0)$. Meanwhile, this ratio $[n_G(0), n_S(0)]^T$ constitutes an eigenvector corresponding to eigenvalue $q_-$. Eventually, we arrived at a condition where the only unknown variables are the front velocity $c$ and decay length $\lambda$.

$$\left[\sqrt{P^2(c,\lambda) + Q(c,\lambda)} + R(c,\lambda)\right]\left[1 - \frac{k_b\chi_{GP}(c,\lambda)M(c,\lambda)}{v_G}\right] = \frac{K_{GS}(c,\lambda)}{v_G}\left[\beta + \frac{k_b\chi_{SP}(c,\lambda)M(c,\lambda)}{v_S}\right] \tag{16}$$

Here $R(c,\lambda) \equiv [\chi_{SP}(c,\lambda)c/\lambda + K_{SG}(c,\lambda)]/2v_S + [\chi_{GP}(c,\lambda)c/\lambda + K_{GS}(c,\lambda)]/2v_G$ and $M(c,\lambda) \equiv [\sqrt{P(c,\lambda)^2 + Q(c,\lambda)} - P(c,\lambda)]/[(\sqrt{P(c,\lambda)^2 + Q(c,\lambda)} - P(c,\lambda))^2 - 1/4\lambda^2]^{3/2}$ for ease of expression. The linear spreading velocity $c_m$ is further determined with the stability condition[49] $dc/d\lambda|_{c=c_m} = 0$ by finding the minimum of $c$ numerically with the MATLAB function 'fmincon' constrained by Eq. (16).

## Phase boundary

The phase boundary (red curve in Fig. 7a and Supplementary Fig. 6) divides the phase diagrams into two regions: one where growth is permitted, and another where no growth can occur. The critical transition is set forth by a positivity requirement of Eq. (12), that $1/\bar{l} + K_{SG}/v_S - K_{GS}/v_G$ on the right-hand side must be greater than zero. Accordingly, one can set $1/\bar{l} = K_{GS}/v_G - K_{SG}/v_S$ in Supplementary Eq. (5) to locate the phase boundary curve.

$$(1-\beta)\left(\frac{K_{GS}}{v_G} - \frac{K_{SG}}{v_S}\right)^2 = k_b\left(\frac{\chi_{GP}}{v_G} + \frac{\chi_{SP}}{v_S}\right) \tag{17}$$

## Traveling wave solution of the one-state model

In the co-moving frame $z \equiv r - ct$, the one-state model can be written as $\partial n(z,l,\theta) = -(c/\bar{v} - \cos\theta/2)n(z,l,\theta)/\lambda$ with the exponential ansatz $e^{-z/\lambda}$, ignoring second-order collision term. Thus, the solution satisfies $n = n(0)e^{-|\nu|l}$ where $|\nu| = c/\bar{v}\lambda - \cos\theta/2\lambda$. Taking the integral over $l$ and $\theta$, along with the boundary condition, we arrive at condition that relates $\lambda$ to $c$:

$$\lambda^2 = \bar{v}^2\left(\frac{c^2}{\bar{v}^2} - \frac{1}{4}\right)^{3/2}/k_b c. \tag{18}$$

When $0 < c < \bar{v}/2$, $\lambda$ is negative hence must be discarded; when $c > \bar{v}/2$, there is always a single positive value of $\lambda$ that satisfies Eq. (18). Consequently, the marginally stable solution $c_m$, which satisfies $dc/d\lambda|_{c=c_m} = 0$, is $c_m = \bar{v}/2$ where $\lambda_m = 0$. The equivalence of the front velocity to the midpoint drift velocity indicates that the one-state expansion is driven exclusively by the elongation of the radial branches within the leading edge. This growth mechanism is predicated on the persistent extension of these subpopulation of branches, which accounts for the sharp density cutoff at the boundary, i.e. the front decay length being zero.

## Numerical solution

For ease of computation, we reduce the system dimensions by taking the 0th and 1st order moment of Eqs. (1)–(3) with respect

to $l$. This doubles the number of equations to 6 but eliminates $l$ as a variable. We discretize radial distance $r$ and radial angle $\theta$ from $(0, 100]$ and $[-\pi, \pi]$ with a $200 \times 100$ mesh. After each time step of 0.05 min, a new set of values are updated using backward Euler method.

## Data availability

The data used in this study have been deposited in Dryad database under accession code https://doi.org/10.5061/dryad.djh9w0w2r.

## Code availability

The code for directed-rod simulation is publicly available at https://github.com/SabyasachiSutradhar/Directed_Rod_Simulation.

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

## Acknowledgements

This work was supported by NIH R01 NS118884 to JH, NSF PHY-2210464 to JH, Pew Charitable Trusts Award 38044 to JH, Yale's Integrated Graduate Program in Physical and Engineering Biology to XO, Fonds de recherches du Québec - Nature et technologies to OT, Harold W. Dodds Fellowship from Princeton University to QY and Flatiron Institute to YT. The Flatiron Institute is a division of the Simons Foundation.

## Author contributions

SSh performed experiments and discovered rebranching; XO, SSh, SSu and OT analyzed data; the initial version of the mean-field model was formulated by OT and YT and the final version, which incorporates length-dependent dynamics and internal transitions, by XO; SSu developed the computational model; JH, XO, SSu, OT, YT and QY contributed theory and computation; and JH, XO and SSu wrote the manuscript with input from SSh, OT, YT and QY. Author contributions are in alphabetical order.

## Competing interests

The authors declare no competing interests.
