## [Transparent Peer Review file · Nature Communications]

Neurons exploit stochastic growth to rapidly and economically build dense dendritic arbors

Corresponding Author: Professor Jonathon Howard

Version 0:

Reviewer comments:

Reviewer #1

(Remarks to the Author)

Ouyang, Sutradhar, et al. developed a, for a non-mathematician, complicated mathematical model that describes how 2D dendrites are built as a minimum spanning tree with radially oriented dendrites to reduce dendritic branch length and to reduce signal propagation times to the cell body. Their model has no free parameters. The manuscript represents an interesting topic and especially the correlation between tip-dynamics and morphology is nice and would be of significance to the field since it will help to gain understanding of how neurons are forming their complex dendritic trees. However, throughout the manuscript I feel like a detailed discussion and comparisons to already existing models for dendrite branching is missing. For example: The link to the Shree et al paper could be better elaborated. Were the imaging data reused from the Sherr et al paper or are those data here newly acquired? It would help readers who are not familiar with growth models and their development to better explain what was the limit in Shree et al and what represents the improvement in this paper. Where are similarities or differences to the models that other labs e.g. the Cuntz lab developed? Also, the c4da neurons are well established models that can be easily altered. Therefore, it would be interesting to look at mutants in which tip dynamics are altered and to check how this would affect the final dendritic morphology in vivo and in the model.

In general, it helps reviewers to add line numbers so that they can precisely refer to a specific part of the text. The intro and the results are not labeled as such. There is not a single line concerning materials and methods in the main text which must be changed (e.g. how was the imaging done, which microscopes were used etc)

Besides Shree et al, Hermann Cuntz also developed a mathematical model that used long-term time-lapse data of growing dendrites which optimizes total wiring and space-filling of c4da neurons. The manuscript is online but has not passed the revision process yet (<https://doi.org/10.1101/2020.07.07.191064>). However, a peer reviewed manuscript on the c3da neurons describes a two-step model that is necessary to describe the c3da neuronal morphology (doi: 10.1016/j.celrep.2022.110746) has also been published. Especially because of the link in between the dendritic tip properties and morphology at least the above mentioned Stüerner et al paper must be discussed and cited. Stüerner et al not only used in vivo time lapse data but their manuscript also included dendritic trees of c3da mutant neurons in which actin regulatory proteins were absent/mutant.

The abstract says "reducing signal propagation" but I could not find where this was calculated only literature that has been cited. Have I overlooked that?

Every figure should explain: What N was used, what does N stand for (number of individuals etc), I am missing which statistical test was used in each figure also sometimes the N (see 2D N = 4) is very small, 24 hours: 8 cells from 3 larvae, 48 hours: 4 cells from 4 larvae, 96 hours: 4 cells from 4..... data are sometimes one neuron per animal, sometimes not. Please make that consistent.

I think that some figures are mislabeled in the text (see Fig 3E)

Introduction:

"In flies alone, over 8,000 neuronal types are distinguished primarily by their shapes in the larval (Winding et al., 2023)".....I don't think this is true for the larva

"can accommodate many synapses or"...synapses should be changed to post-synaptic sites

"Evidence comes from live imaging in several experimental systems"please add Stüerner et al doi: 10.1016/j.celrep.2022.110746

„that receive mechanical input“.....they are nociceptors reacting to harsh mechanical stimuli, light, heat.....please add that information and references

“While these cells do not receive synaptic input” change cells to dendrites

“Over the five days of larval development” The larva develops in three-four days, everything before is embryonic development

“class IV arbors grow from 100 μm to 500 μm in diameter“ there is already data on that and this should be cited Peng et al. doi: 10.1242/jcs.174771. or Ziegler et al. doi: 10.1016/j.celrep.2017.11.069

Results:

“class IV neurons at 24 (13 cells), 48 (5 cells), and 96 (9 cells) hours AEL” I prefer a similar N (or at least more than 5) and I think you should apply a statistical test to compare the trees between the larval stages or a way that compares the basic shape of the curves independent of their distance to the Soma

Figure 1:

I think it would make sense to split that figure into two 1A-G and H-O

A A8/A9 A9 is missing

F and G axes are not clearly labeled, label belongs to the left side in F or move the y-axis to the right

F and G One point of comparing different developmental stages is to test if there is a difference or not, but I do not see any statistical test here

“Branching, growth, and retraction occur on short timescales” that has been shown multiple times already, maybe not as detailed as here but the according literature should be cited

Figure 2:

Figure 2D I think the unit (beta) should be explained a bit better in the main text or a direct comparison of branching at a random site and rebranching could be added in the figure

Figure 3E:

The graph is hard to read, Class IV cells (Figure 3E, green circles), agent-based simulations of class IV cells (Figure 3E, blue squares)... Figure 3E, red triangles Where are those or is the figure mislabeled? Please check throughout the manuscript if all figures are cited accordingly

Figure 6:

A: to which stage does the tree corresponds to

Discussion:

(see references in the Introduction) ->please add those to the discussion

Such non-cell autonomous mechanisms are important to restrict class IV dendrites to their segments during growth (Parrish et al., 2009). -> this part should be more detailed; I fear people not working on c4da neurons/tiling would not understand it easily.

Extracellular cues have also been shown to be important for the dendrites of Purkinje cells (Joo et al., 2014). -> also, diet plays a role in dendrite branching (high protein for example)

However, the mechanisms by which these perturbations alter dendritic geometry (e.g., branch

length and density, arbor size) are not known. -> is this true? There should be a lot of literature describing how alter cytoskeletal protein composition affects dendritic morphology by altering tip growth dynamics, especially for the c4da neurons which were used extensively as a model to study the effect of cytoskeletal regulators.

Reviewer #2

(Remarks to the Author)

Overall, the manuscript is well written and makes a strong contribution to the research on dendritic branching dynamics. The authors extend their previous published results (Shree et al 2022), and make a comprehensive study on stochastic modelling of dendrite arbor dynamics. They developed a mean-field model to study the statistical behavior of arbor branching and analytically studied its steady state solution as well as time dependent solutions of its corresponding 1-state model reduction. They also simulate the steady state density using directed rod stochastic process. Their model recaptures several key morphological properties of class IV Drosophila sensory dendrites, including branching length distribution, the scaling between dendrite number and length density, and the tight spacing of dendritic meshwork. They consider the dense meshwork is built with minimal total branch length and stochastic growth accelerates the expansion rate of the arbor.

Below are several detailed comments and questions.

- 1、 P5, Fig2, when presenting sem+sd from experimental data, it would be good to give sample size n
- 2、 The model assume branch only occurs at the tip ? is there any experimental evident ? could it occur at internal branch ?
- 3、 P6, Table 1, alpha is about 1.4, however, in Table 2 caption is says alpha=0.75
- 4、 P7-8, the mean-field model equation 1-4, eg. Eq 1 has the term $T \rightarrow S$, how about the case $G \rightarrow I$, also cases $S \rightarrow I$ in Eq (2), $P \rightarrow I$ in Eq(3), $I \rightarrow X$ in Eq (4). are those missing ?
- 5、 According to Fig2A, the authors assume when collision occurs, the branch disappear. In fig2A, this occurs within 5mins, it would be interesting to see how branch disappear, does it shrink faster than others in shrink state ?
- 6、 P12, line2, should that be Fig 4E ?
- 7、 In Cuntz 2012 PNAS, they have considered dendrite as optimal rewiring minimizing the total length. It might be worth to discuss how the results compared to preciously dendritic optimal models. What's new we can learn from the mean-field model compared to preciously dendritic optimal models?

8、 It might be better to move Box 1 to Online Methods.

9、 When referring to Online Method in main text, it might be better to refer more explicitly (e.g. which section of Online Method) so that readers can find easily.

10、 p14, The authors argue that diffusion (four fold reduction) compared to drift (37% decrease) play a major role in driving arbor expansion. Could that be understood simply from that diffusion rate is of order 0.5 while drift is an order of 0.03. Is the statement is still true using different choice of D and v along the boundary curve in Fig5G, say high v and low D ?

11、 P 15, line -3 'The model can then be used to extrapolate molecular perturbations of tip dynamics to those of the whole arbor.' How to extrapolate molecular perturbations ? please discuss this in detail.

Reviewer #3

(Remarks to the Author)

The manuscript presents a modelling approach for dendritic branching, elongation, and retraction based on stochastic tip dynamics. The authors use a mean-field approximation to describe the relationship between dendrite growth/shrinkage and their average lengths and densities. Key predictions from the model include:

- 1) Exponential distribution of branch lengths.
- 2) Parabolic scaling between dendrite number and length densities.
- 3) Dense dendritic meshwork with minimal total branch length, radially oriented branches, and accelerated expansion of the arbor.

These findings are consistent with experimental observations in class IV *Drosophila* sensory dendrites. The model highlights the efficiency of stochastic growth mechanisms in producing space-filling dendritic arbors and may offer a general theoretical framework for exploring dendritic morphogenesis.

The manuscript addresses an important problem in dendritic differentiation and proposes a model that could potentially generalize to other systems. However, the novelty and broader impact of this study are not immediately apparent. If the focus of the paper is on the methodological development, then the authors should demonstrate the utility of the mean-field approximation across other cell types or biological contexts.

While the model builds on stochastic tip dynamics during dendrite differentiation, the manuscript lacks a clear articulation of the central research question. It is unclear what specific gap in knowledge this study aims to fill, especially considering the existing body of work on dendritic growth models. The rationale for the use of a mean-field approximation also needs clarification. The manuscript should explain why this approach provides new insights that are not accessible through existing models.

The functional insights drawn from the model (e.g., minimal branch length, radial orientation, and rapid space-filling) are interesting, but it is unclear how they advance the field beyond what was already known. Highlighting clear functional insights or potential experimental applications could enhance the significance of the study

Additionally, while the model provides a framework for space-filling dendrites, it is not clear if it can be applied to neurons or other cell types with non-space-filling morphologies. Space-filling dendrites (like class IV *Drosophila* dendrites) have distinct growth strategies that are not necessarily present in retinal ganglion cells, cortical pyramidal neurons, or other neurons. It is important for the authors to discuss whether this model can be applied to cells that do not exhibit dense arborization.

One significant issue that warrants clarification is how the current model reconciles or interacts with experimental findings from previous works. For instance:

1) Yoong et al. (2020, *Neuron*) and Baltruschat et al. (2019, bioRxiv) propose a different growth process where primary long branches (or a "backbone") first innervate vacant space, and subsequent branching fills the remaining space. This "backbone-first" process contrasts with the purely stochastic growth approach presented in this manuscript. The authors should explicitly discuss how their model can incorporate or explain this hierarchical process.

2) Can this model be adapted to capture a two-phase process where primary branches are first established, and secondary branches subsequently emerge from these backbones? If the mean-field approach is inherently limited to treating all branches as equivalent, this limitation should be explicitly stated.

3) If the authors believe their model does capture these experimental results, they should clearly explain how this happens and highlight the specific elements of the model that account for the "backbone-first" growth observed experimentally.

4) If this model represents an alternative explanation, the authors should provide a justification for why the current approach provides unique insights or is advantageous relative to prior work.

The authors' model assumes that the dendritic arbor grows as a 2D planar system, but this assumption is not sufficiently discussed. Since some biological neurons grow in 3D, it would be beneficial to discuss the potential limitations of this assumption and how it impacts the generalizability of the model. Additionally, the authors should highlight the extent to which their model can generalize to dendrites that are not constrained to 2D space. Finally, while the model focuses on class

IV Drosophila dendrites, it is not clear if it can generalize to other neuronal types. The authors should explain how the model could be extended to other non-space-filling neuronal types (e.g., cortical pyramidal neurons) or address why it is restricted to space-filling morphologies.

Data and Methodology:

The authors claim that the mean-field approximation offers clear advantages over branch-tracking models, which track the position, state, and hierarchy of individual dendritic branches. This is a compelling argument, especially in the context of the computational demands of tracking thousands of dynamic branches. However, with the rise of computer vision techniques that now allow automated branch tracking "for free", it is important for the authors to clearly articulate why the mean-field approach remains advantageous. The authors should highlight the computational, conceptual, or interpretive benefits of mean-field modeling that remain valuable even in the presence of automated tracking methods. This is key taking into account the previous work from the lab (Shree et al, 2022). At this point, it is not clear how the current manuscript differentiates itself from Shree and colleagues, 2022.

If the key advantage is computational efficiency, this should be quantified. If the advantage is conceptual (e.g., a more general view of dendritic growth dynamics), this should be clearly stated in the introduction or discussion.

The authors use a mean-field approach to describe the growth dynamics of dendritic tips. This approach simplifies the complexity of dendritic growth by focusing on average properties rather than individual branch states. However, it is critical to address the following issues:

- 1) Clarity of the model's assumptions: The assumption that the dendritic arbor grows as a 2D planar system is stated but not sufficiently discussed. Since most biological neurons grow in 3D, it would be beneficial to discuss the potential limitations of this assumption and how it impacts the generalizability of the model.
- 2) Description of simulation procedures: There is reliance on simulations (both directed-rod and mean-field) without clear, detailed descriptions of the simulation procedure. Readers may be left wondering about the exact steps used to generate the data. Providing a clear, step-by-step explanation of how the simulations are performed, including key parameters, would address this issue.
- 3) Code availability: No code is provided to reproduce the figures. This limits the reproducibility and transparency of the study. Making the code available would enhance transparency, allow for validation of the model, and facilitate future work that builds on this model.
- 4) Comparison with backbone-first models: Experimental studies, such as Yoong et al. (2020, Neuron) and Baltruschat et al. (2019, bioRxiv), propose a "backbone-first" process for dendritic growth. The authors should explain how their model accommodates, incorporates, or contradicts this process.
- 5) Biological Implications: The theoretical aspects of branching are well-covered, but the biological relevance is underexplored. For example, the authors should discuss how the model helps explain unique features of neuronal types beyond class IV Drosophila cells.
- 6) Potential confounding factors: External factors like the impact of the extracellular matrix, or guidance cues are not fully explored. Addressing these confounding factors would provide a more comprehensive picture of dendritic morphogenesis.

Suggested Improvements

- 1) Clarify the research question: The authors should articulate the specific problem or gap in knowledge that this study addresses earlier in the manuscript.
- 2) Highlight the model's novelty: Clearly define how the mean-field approach provides new insights that are not possible with existing models.
- 3) Include simulation details: Provide a clear, step-by-step explanation of the directed-rod and mean-field simulation procedures, including all parameters.
- 4) Address 2D vs 3D assumptions: Discuss the impact of the assumption that dendritic growth is constrained to 2D and its limitations.
- 5) Provide the code for reproducibility: Make the code publicly available to enable others to reproduce the figures and validate the findings.
- 6) Address potential confounding factors: Discuss how the model might be influenced by extracellular matrix, or guidance cues, and clarify how these elements could be incorporated (or not) into the model.
- 7) Address backbone-first models: Discuss the model's relationship with the "backbone-first" growth process proposed by

Yoong et al. (2020, Neuron) and Baltruschat et al. (2019, bioRxiv).

8) Adjust text-to-equation ratio: Provide more explanatory text around the key equations, especially for readers unfamiliar with modelling.

Version 1:

Reviewer comments:

Reviewer #1

(Remarks to the Author)

The submitted version of the manuscript is substantially improved and I would suggest accepting it. My only critical suggestion is to mention whether the data have been previously analyzed for normal distribution, or which test was used, because only normally distributed data should be analyzed with a one-way ANOVA test.

(Remarks on code availability)

Reviewer #2

(Remarks to the Author)

The manuscript is well written and makes a strong contribution to the research on dendritic branching dynamics. The authors make a comprehensive study on stochastic modelling of dendrite arbor dynamics, developing a sophisticated mean-field model to study the statistical behavior of arbor branching. Their modelling results recapture several key morphological properties of class IV *Drosophila* sensory dendrites. Modelling, calculations as well as experiments are well detailed in methods. The revised manuscript together with the response answered all my questions and I consider the manuscript is worth to be published in Nature Communications.

(Remarks on code availability)

Point-by-point

Point-by-point response to reviewers' comments on
"Neurons exploit stochastic growth to rapidly and economically build dense radially
oriented dendritic arbors"
By Ouyang, Sutradhar et al.
Nature Communications

REVIEWER COMMENTS

General response to the reviewers

We thank the reviewers for their detailed comments and suggestions and hope they have led to an improved manuscript.

There were two main issues that were raised by two or more of the Reviewers.

1. Motivation, novelty and gap in knowledge, new insights, discussion of other models (reviewers 1 and 3)

First, we apologize to the reviewers for the lack of an Introduction and motivation for the mean-field model. We had submitted it to *Nature Physics* and simply pushed the button to transfer to *Nature Communication* when *Nature Physics* declined to review it. This was a mistake because the two journals have quite different formats. The former has a stringent word limit, no Introduction, and limited Discussion.

In the new Introduction, we have motivated the mean-field model and argued how it goes beyond the agent-based models of Shree et al. (2022) and many others. One important feature of the mean-field model is its conceptual simplification and the ability to obtain analytic expressions for how the dendrite geometry (lengths, densities, orientations, etc.) depends on the tip parameters. Secondly, and as importantly, we show that the mean-field model can test the role of topology in dendrite morphology. This is one of the "gaps in knowledge". Both Reviewer 1 and 3 were concerned about our failure to consider other models properly, such as the two-step model (Stürner et al., 2022) and backbone-first hierarchical models (Baltruschat et al., 2020; Ferreira Castro et al., 2020a). The models in these papers, which are now discussed in the Introduction, are topological models in the sense that the branching depends on the position within the tree structure. The nice feature of the mean-field model is that branching is treated as nucleation, and there is no tree structure (conceptually, it is a collection of isolated branches). Therefore, the ability of the mean-field model to predict the geometric properties of class IV cells shows that hierarchical branching mechanisms are not necessary, at least for class IV cells. We think that this is an important new insight. In the Discussion, we come back to the Stürner and Baltruschat (and other) papers, because it is clear that topology does matter for class I and class III cells.

2. Mathematical and technical issues (reviewers 1 and 2)

Again, we apologize because the manuscript was aimed at a *Nature Physics* readership. In the revised manuscript, we have considerably simplified and shortened the mathematical equations in the Box (though we prefer to keep it in the main text) by moving the more complex theory relating to the internal-terminal interconversions to the

Point-by-point

end of Supplementary Information. In addition, we have included a new section (“Three-state mean-field model”) describing the model in non-mathematical terms. Finally, we have reduced the number of times that we refer to the equations.

Reviewer #1 (Remarks to the Author):

Ouyang, Sutradhar, et al. developed a, for a non-mathematician, complicated mathematical model ...

As described above, we have simplified the model in the text and, as much as possible, described it in non-mathematical terms.

... that describes how 2D dendrites are built as a minimum spanning tree with radially oriented dendrites to reduce dendritic branch length and to reduce signal propagation times to the cell body. Their model has no free parameters. The manuscript represents an interesting topic and especially the correlation between tip-dynamics and morphology is nice and would be of significance to the field since it will help to gain understanding of how neurons are forming their complex dendritic trees.

However, throughout the manuscript I feel like a detailed discussion and comparisons to already existing models for dendrite branching is missing. For example: The link to the Shree et al paper could be better elaborated.

As stated in the “General Response” section above, we have articulated in the Introduction how our mean-field model differs from existing models including the Shree et al. model.

Were the imaging data reused from the Shree et al paper or are those data here newly acquired?

Much of the data is from the Shree et al. paper (most of the rates in Table 1), though we have increased the number of animals and cells in Figures as requested. The important finding of rebranching was not in the Shree paper. We make the relationship clearer in the results by saying in the Results:

"As previously reported by Shree et al (2022), after formation of a new branch, tips grow, shrink, and pause, and, following collision with the shaft of another dendrite, they retract and disappear (Figure 3A,B). In addition to these behaviors, we discovered that branches sometimes reappear at the sites of disappearance (Figure 3C)."

Most of the analysis is new: density (Figure 1), orientation (Figure 2), rebranching (Figure 3), number and length scaling (Figure 5), space-filling efficiency (Figure 5), front decay length (Figure 6), and detailed orientational analysis (Figure 8).

It would help readers who are not familiar with growth models and their development to better explain what was the limit in Shree et al and what represents the improvement in this paper. Where are similarities or differences to the models that other labs e.g. the Cunz lab developed?

Point-by-point

In the new Introduction, we have explained the limitations and differences between agent-based models (such as Shree et al) and mean-field models, and the advantages of the latter (simplicity, ability to obtain analytic solutions). Also, we point out important conceptual differences, such as the fact that the mean-field model has no network structure or topology so can test hierarchical models.

In the new Introduction, we differentiate between the growth models (agent-based and mean-field) that are bottom-up, and minimal spanning tree models (and their elaboration by Cuntz and co-workers), which are top-down. In these latter models, there is pre-patterning followed by deterministic rules to create the network. The agent-based and mean-field models are, therefore, complementary to the minimal-spanning-tree models.

Also, the c4da neurons are well established models that can be easily altered. Therefore, it would be interesting to look at mutants in which tip dynamics are altered and to check how this would affect the final dendritic morphology in vivo and in the model.

Analysis of mutants is in ongoing work. But in a section in the Discussion (Insights into Molecular Phenotypes), we indicate how our model can be used to understand mutant phenotypes.

In general, it helps reviewers to add line numbers so that they can precisely refer to a specific part of the text.

Line numbers have been added.

The intro and the results are not labeled as such.

The Introduction and Results sections are now labeled.

There is not a single line concerning materials and methods in the main text which must be changed (e.g. how was the imaging done, which microscopes were used etc).

Some methodological details about the imaging and image analysis are in the Figure Legends. Detailed experimental methods are now included in the newly added Methods section.

Besides Shree et al, Hermann Cuntz also developed a mathematical model that used long-term time-lapse data of growing dendrites which optimizes total wiring and space-filling of c4da neurons. The manuscript is online but has not passed the revision process yet (<https://doi.org/10.1101/2020.07.07.191064>). However, a peer reviewed manuscript on the c3da neurons describes a two-step model that is necessary to describe the c3da neuronal morphology (doi: 10.1016/j.celrep.2022.110746) has also been published. Especially because of the link in between the dendritic tip properties and morphology at least the above mentioned Stuermer et al paper must be discussed and cited. Stuermer et al not only used in vivo time lapse data but their manuscript also

Point-by-point

included dendritic trees of c3da mutant neurons in which actin regulatory proteins were absent/mutant.

In the new Introduction, we discuss and reference both these papers (as well as the important Ferreira et al. 2020 paper). We return to them in the Discussion. Part of the motivation of the mean-field model is a way of testing whether two-step and other hierarchical models (“space-filling growth tree”) are necessary to generate the morphology of class IV cells (they are not).

In the Results, we draw attention to the fact that the parameter theta (θ) in Baltruschat et al. (2020, bioRxiv) is related to our mesh size parameter and that the concepts of efficiency of space-filling are similar in the two papers. However, it must be stressed that the models are very different: our model is bottom-up (with growth rules), while the Baltruschat et al. model is top-down (from a predetermined template). Thus, the results are complementary.

The abstract says “reducing signal propagation,” but I could not find where this was calculated, only literature that has been cited. Have I overlooked that?

We interpret radial orientation as a mechanism to reduce the distance between excitation points and the cell body (along the branches). We were motivated by the Cuntz balancing factor paper (2010), where distance to the soma is one of the constraints of the minimization. The Figure below shows that the higher the balancing factor, the shorter the propagation times (which are proportional to the distance to the soma, left panel) and that this correlates with radial orientation (inset, right panel). The mean path distance from the excitation point to the root exhibits a strong negative correlation with orientation, indicating that as the trees became more radial (higher balancing factor), the path distance to the root decreases. Thus, the more radial the branches, the shorter the signal travel times to reach the soma.

Figure: Effect of radial orientation of signal propagation distances. 500 minimum spanning trees were generated by randomly distributing 250 points within a 100×100 micron box, each corresponding to different balancing factors (bf). **A** Two such trees with $bf=0$ and $bf=1$ are shown. For each tree, a random node was selected within an annular region with an average radius 45 microns (2 microns wide) and designated as the excitation point (red circles on the trees). The path distance along the branches to the root was then computed and plotted against the balancing factor. As the balancing factor increased, the path distance to the root decreased. **B** Additionally, the mean radial orientation was determined by calculating the angle between the radial direction and each branch, then averaging the results. A strong linear correlation was observed between the balancing factor and the mean radial orientation as shown in the Inset.

Point-by-point

Every figure should explain: What N was used, what does N stand for (number of individuals etc),

In the revised manuscript, we have stated the total number of cells and the total number of animals in the Figure legends.

I am missing which statistical test was used in each figure ...

We used ordinary ANOVA with Tukey's correction for multiple comparisons to determine the significance and added the results of the statistical test to the figures. The test is explained in the Methods.

... also sometimes the N (see 2D N = 4) is very small, 24 hours: 8 cells from 3 larvae, 48 hours: 4 cells from 4 larvae, 96 hours: 4 cells from 4..... data are sometimes one neuron per animal, sometimes not. Please make that consistent.

We have increased the number of animals to a minimum of 6. Because the neurons in younger larvae are smaller, the number of cells in a field of view is larger, and so we tend to have more cells per animal in younger larvae. However, the important thing is to have larger numbers of animals (to control for animal-to-animal variation).

I think that some figures are mislabeled in the text (see Fog 3E)

Figure labels are fixed.

Introduction:

"In flies alone, over 8,000 neuronal types are distinguished primarily by their shapes in the larval (Winding et al., 2023)".....I don't think this is true for the larva

The reviewer is correct - this is not true for the larva brain. We have rephrased to make it clear that we are referring to larval AND adult brains.

"can accommodate many synapses or"...synapses should be changed to post-synaptic sites

Changed to post-synaptic sites.

"Evidence comes from live imaging in several experimental systems"please add Stürner et al doi: 10.1016/j.celrep.2022.110746

We have cited the Stürner et al paper in the Introduction of the revised manuscript, as indicated above.

„that receive mechanical input“.....they are nociceptors reacting to harsh mechanical stimuli, light, heat.....please add that information and references

Point-by-point

We have changed the sentence to add more references to the modalities. “The highly branched arbors of *Drosophila* class IV da nociceptors sense noxious stimuli, including heat (Tracey et al., 2003), ultra-violet light (Xiang et al., 2010), and harsh mechanical stimuli (Hwang et al., 2007) over a large area of the larval surface (Figure 1A,B) (Jan and Jan, 2010; Shree et al., 2022).”

“While these cells do not receive synaptic input” change cells to dendrites

“Cells” are changed to “dendrites”.

“Over the five days of larval development” The larva develops in three-four days, everything before is embryonic development

At our temperature of 25 degrees L1 starts at 24hrs AEL and the larvae pupate around 144 hours AEL, which is after 5 days of larval development. The temperature is stated in the Methods. At 28 degrees, development is about a day faster.

“class IV arbors grow from 100 μm to 500 μm in diameter“ there is already data on that and this should be cited Peng et al. doi: 10.1242/jcs.174771. or Ziegler et al. doi: 10.1016/j.celrep.2017.11.069

We cited the Grueber et al. (2002) paper, which is earlier than these. We cite these papers in the Discussion in the context for perturbations of morphology.

Results:

“class IV neurons at 24 (13 cells), 48 (5 cells), and 96 (9 cells) hours AEL” I prefer a similar N (or at least more than 5)

We have increased the number of cells at 48 hours AEL to 12, to more closely match the numbers at other ages.

and I think you should apply a statistical test to compare the trees between the larval stages or a way that compares the basic shape of the curves independent of their distance to the Soma

We now state in the text that the steady-state densities decrease and the arbor diameters increase over development, and that these changes are statistically significant (Figure 5A,B and legends). The shape of the expanding front (Figure 1F-G and Figure 6B) can be characterized by the front decay length, which is plotted in Figure 6F together with the statistical significance (with details in the Figure 6 legend).

Figure 1:

I think it would make sense to split that figure into two 1A-G and H-O

We have followed the reviewer's advice.

Point-by-point

A A8/A9 A9 is missing

Fixed.

F and G axes are not clearly labeled, label belongs to the left side in F or move the y-axis to the right

Fixed.

F and G One point of comparing different developmental stages is to test if there is a difference or not, but I do not see any statistical test here

The dark lines in F and G are the means and the light lines are the SEMs. We have added this to the figure legend. We give the statistical analysis in Figures 5B and 6B and give details of the tests in Figure legends.

“Branching, growth, and retraction occur on short timescales” that has been shown multiple times already, maybe not as detailed as here but the according literature should be cited

The Introduction includes citations to earlier studies in multiple organisms. We think Guo et al., 1999, is the first reference on dynamics in Class IV cells, but we would be happy to be corrected.

Figure 2:

Figure 2D I think the unit (beta) should be explained a bit better in the main text or a direct comparison of branching at a random site and rebranching could be added in the figure

We have now more clearly defined beta in the text.

Note that β is a dimensionless parameter that quantifies the probability of a rebranching event occurring. To avoid confusion, we moved β out from the bracket in the y-axis label of Figure 3D. We measured β by counting the reappearance frequency of branches after their disappearances. See details in Methods, “Rebranching probability” section.

Figure 3E:

The graph is hard to read, Class IV cells (Figure 3E, green circles), agent-based simulations of class IV cells (Figure 3E, blue squares)... Figure 3E, red triangles Where are those or is the figure mislabeled? Please check throughout the manuscript if all figures are cited accordingly

We corrected the figure labels. This is now Figure 5E in the revised manuscript. Sorry for the confusion. We also adjusted the colors and made the figure more legible.

Figure 6:

Point-by-point

A: to which stage does the tree corresponds to

The tree is from a cell of age 48 hours AEL. This is now stated in the figure legend.

Discussion:

(see references in the Introduction) -> please add those to the discussion

Done.

Such non-cell autonomous mechanisms are important to restrict class IV dendrites to their segments during growth (Parrish et al., 2009). -> this part should be more detailed; I fear people not working on c4da neurons/tiling would not understand it easily. Extracellular cues have also been shown to be important for the dendrites of Purkinje cells (Joo et al., 2014). -> also, diet plays a role in dendrite branching (high protein for example)

We have expanded the Discussion to include these non-cell autonomous mechanisms and external cues.

However, the mechanisms by which these perturbations alter dendritic geometry (e.g., branch length and density, arbor size) are not known. -> is this true? There should be a lot of literature describing how alter cytoskeletal protein composition affects dendritic morphology by altering tip growth dynamics, especially for the c4da neurons which were used extensively as a model to study the effect of cytoskeletal regulators.

There is a large literature on how genetic and other (diet) perturbations alter class 4 morphologies, and in the Discussion (“Insights into molecular phenotypes” section), we indicate how the models can help to understand the phenotypes.

Reviewer #2 (Remarks to the Author):

Overall, the manuscript is well written and makes a strong contribution to the research on dendritic branching dynamics. The authors extend their previous published results (Shree et al 2022), and make a comprehensive study on stochastic modelling of dendrite arbor dynamics. They developed a mean-field model to study the statistical behavior of arbor branching and analytically studied its steady state solution as well as time dependent solutions of its corresponding 1-state model reduction. They also simulate the steady state density using directed rod stochastic process. Their model recaptures several key morphological properties of class IV Drosophila sensory dendrites, including branching length distribution, the scaling between dendrite number and length density, and the tight spacing of dendritic meshwork. They consider the dense meshwork is built with minimal total branch length and stochastic growth accelerates the expansion rate of the arbor.

Below are several detailed comments and questions.

1、 P5, Fig2, when presenting sem+sd from experimental data, it would be good to give sample size n

Point-by-point

Sample sizes are given in the figure legends.

2、 The model assume branch only occurs at the tip ? is there any experimental evident ? could it occur at internal branch ?

Branching is observed on both terminal and internal branches. This is now stated in the text. In the mean-field model, branching is a nucleation process whose rate is proportional to the length density of branches (as would be the case for real branching). Nucleation is proportional to the total branch length density, $\rho = \rho_T + \rho_I$, which incorporates both terminal and internal branch densities.

3、 P6, Table 1, alpha is about 1.4, however, in Table 2 caption is says alpha=0.75

The α in Table 1 is fitted from the 3-state directed-rod simulations, where tips transition between the 3 states. The α in Table 2 is fitted from the 1-state directed-rod simulations, where branches grow consistently. The value of α in three-state simulations depends on the effective diffusion coefficient (see Supplementary Figure S3C).

4、 P7-8, the mean-field model equation 1-4, eg. Eq 1 has the term $T_I \rightarrow S$ (the reviewer means $T_I \rightarrow G$ here), how about the case $G \rightarrow I$, also cases $S \rightarrow I$ in Eq (2), $P \rightarrow I$ in Eq(3), $I \rightarrow X$ in Eq (4). are those missing ?

First, we have moved this part of the theory to the Supplementary Information (“Full model including terminal-internal dendrite interconversion”), so Equations 1-3 only apply to terminal branches. This simplifies the main text, as requested by the Editor.

Second, the term $T_I \rightarrow G$ is the net conversion from internal branches to growing terminal branches, as described in Supplementary Information Equations S16-18. It includes both loss and gain of growing terminal dendrites resulting from branching. Likewise, for the shrinking and paused terminal dendrites.

5、 According to Fig2A, the authors assume when collision occurs, the branch disappear. In fig2A, this occurs within 5mins, it would be interesting to see how branch disappear, does it shrink faster than others in shrink state ?

In our previous paper (Shree, et al., 2022), post-collision tip dynamics was measured, see parameters in Table 1 (Shree, et al., 2022). The takeaway is that the tip velocities are more or less the same before and after collisions, but the transition rates change significantly so that branches shrink more persistently after collision, resulting in branch disappearance in about 10 minutes.

6、 P12, line2, should that be Fig 4E ?

Corrected to Figure 4E.

Point-by-point

7、 In Cuntz 2012 PNAS, they have considered dendrite as optimal rewiring minimizing the total length. It might be worth to discuss how the results compared to preciously dendritic optimal models. What's new we can learn from the mean-field model compared to preciously dendritic optimal models?

1. Cuntz et al. (2012) show that "optimal wiring" (i.e., dendrites simulated with the balancing factor model) gives a scaling law between dendrite length and dendrite number, with an exponent of $2/3$ for 3D and $1/2$ for 2D. Importantly, we show that the parabolic scaling in 2D follows from stochastic branching (without making assumptions about optimal wiring).

2. The scaling factor of $1/2$ (in 2D) follows in a very straightforward way from our one-state mean-field model. For the 3-state model, the parabolic relation is only approximate, though quite close to a parabola.

3. In addition to being based on the observed stochastic growth, the mean-field model makes many other predictions, including density and arbor expansion speed, which the Cuntz model does not.

We make these points in the paper.

8、 It might be better to move Box 1 to Online Methods.

We prefer to keep the Box in the main text but have simplified it by moving the internal-terminal theory to the Supplementary Information.

9、 When referring to Online Method in main text, it might be better to refer more explicitly (e.g. which section of Online Method) so that readers can find easily.

We have now done this.

10、 p14, The authors argue that diffusion (four fold reduction) compared to drift (37% decrease) play a major role in driving arbor expansion. Could that be understood simply from that diffusion rate is of order 0.5 while drift is an order of 0.03. Is the statement is still true using different choice of D and v along the boundary curve in Fig5G, say high v and low D ?

The 4-fold increase and 37% decrease are for the measured values of D and v . However, as the reviewer intimates, the changes depend on the values of D and v (through the dimensionless Peclet number, Supplementary Figure S3D). For example, when D is very small (one-state model), the growth depends only on v . On the other hand, when D is very large, there is arbor growth even when v is negative.

On the boundary curve (red curve in Figure 5G, which is now defined in the Figure legend), a decrease in v or D leads to no growth (the arbor will shrink and disappear).

We have clarified this in the text.

Point-by-point

11、 P 15, line -3 ‘The model can then be used to extrapolate molecular perturbations of tip dynamics to those of the whole arbor.’ How to extrapolate molecular perturbations ? please discuss this in detail.

We have added a new section in the Discussion entitled: “Insights into molecular phenotypes”. We argue that our work establishes a morphogenetic “pathway” that causally links tip dynamics to arbor geometry. We show that the model can then be used to extrapolate molecular perturbations of tip dynamics to those of the whole arbor. Using the one-state model for simplicity, we show how proteins that influence tip growth rate or change the rate of spontaneous conversion from the growing to the shrinking state are expected to change dendrite density and length and the rate of growth of the arbor.

Reviewer #3 (Remarks to the Author):

The manuscript presents a modelling approach for dendritic branching, elongation, and retraction based on stochastic tip dynamics. The authors use a mean-field approximation to describe the relationship between dendrite growth/shrinkage and their average lengths and densities. Key predictions from the model include:

- 1) Exponential distribution of branch lengths.
- 2) Parabolic scaling between dendrite number and length densities.
- 3) Dense dendritic meshwork with minimal total branch length, radially oriented branches, and accelerated expansion of the arbor.

These findings are consistent with experimental observations in class IV *Drosophila* sensory dendrites. The model highlights the efficiency of stochastic growth mechanisms in producing space-filling dendritic arbors and may offer a general theoretical framework for exploring dendritic morphogenesis.

The manuscript addresses an important problem in dendritic differentiation and proposes a model that could potentially generalize to other systems.

Thank you.

However, the novelty and broader impact of this study are not immediately apparent. If the focus of the paper is on the methodological development, then the authors should demonstrate the utility of the mean-field approximation across other cell types or biological contexts. While the model builds on stochastic tip dynamics during dendrite differentiation, the manuscript lacks a clear articulation of the central research question. It is unclear what specific gap in knowledge this study aims to fill, especially considering the existing body of work on dendritic growth models. The rationale for the use of a mean-field approximation also needs clarification. The manuscript should explain why this approach provides new insights that are not accessible through existing models.

We have revised the Introduction to clarify the research question (i.e., the “gap in knowledge”). In our view, a key question that the mean-field model addresses is whether the growth of class IV-cell dendrites depends on the topology of the dendrite network. There are several alternative models in which topology plays a key role in morphogenesis. This includes the “two-step” models mentioned by Reviewer 1 and the

Point-by-point

“back-bone first” model mentioned by Reviewer 3. In these models the branching rules differ depending on the hierarchy of the branches within the arbor (which is a topological property). Because the mean-field model has no tree structure, there is no topology. So, if the mean-field model can accurately predict arbor properties, then these properties cannot depend on the topology, at least in class IV cells. Thus, the mean-field model serves as a null hypothesis for testing the role of topology in morphogenesis. This is an important point because topology is difficult to address, and it is not clear the role topology plays in the agent-based model of Shree et al., which generates hierarchical morphologies.

The mean-field model has several additional advantages over agent-based models. The main one is that its simplification allows analytic solutions, which can then identify key parameters that determine morphology.

These arguments are now contained in the new Introduction, which sets up the gaps in knowledge and provides justification for the mean-field approach.

The functional insights drawn from the model (e.g., minimal branch length, radial orientation, and rapid space-filling) are interesting, but it is unclear how they advance the field beyond what was already known. Highlighting clear functional insights or potential experimental applications could enhance the significance of the study

In addition to testing the role of topology, the model also shows that functional properties (branch length, orientation, optimal space-filling, parabolic length-density scaling) emerge from the local growth rules. We have added a new section in the Discussion entitled “Insights into molecular phenotypes,” where we argue that our model can predict how alterations in tip dynamics give morphological phenotypes. This gives insights into the etiology of mutations.

Additionally, while the model provides a framework for space-filling dendrites, it is not clear if it can be applied to neurons or other cell types with non-space-filling morphologies. Space-filling dendrites (like class IV *Drosophila* dendrites) have distinct growth strategies that are not necessarily present in retinal ganglion cells, cortical pyramidal neurons, or other neurons. It is important for the authors to discuss whether this model can be applied to cells that do not exhibit dense arborization.

Our approach has been to focus on one cell type (*Drosophila* class IV cells) and test the models quantitatively against experimental data. The applicability of the model to other cells is an important question. We devote a section in the Discussion (“Caveats, limitations, and generalizations”) to discussing which cells can and cannot be modeled with our approach and how modifications to the mean-field model (such as adding a two-step mechanism) or generalizing collisions to 3D could increase the number of cell types that the mean-field model could apply to.

One significant issue that warrants clarification is how the current model reconciles or interacts with experimental findings from previous works. For instance:

Point-by-point

1) Yoong et al. (2020, Neuron) and Baltruschat et al. (2019, bioRxiv) propose a different growth process where primary long branches (or a "backbone") first innervate vacant space, and subsequent branching fills the remaining space. This "backbone-first" process contrasts with the purely stochastic growth approach presented in this manuscript. The authors should explicitly discuss how their model can incorporate or explain this hierarchical process.

The first point is that the mean-field model does not have a hierarchy of branches, yet it still predicts many of the morphological features of class IV cells. Thus, there is no need to invoke a backbone-first, infilling process for class IV cells: infilling can occur through stochastic branching and growth.

Other cells clearly have hierarchical branching. We have added a paragraph in the Discussion:

"Hierarchical branching is certainly important in other cells. *Drosophila* Class III da neurons have actin-rich branchlets along the backbone of their dendrite (Liao et al., 2023; Stürner et al., 2022; Tsubouchi et al., 2012) and grow by a back-bone first mechanism (Stürner et al., 2022). The branchlets may be analogous to the dendritic spines of vertebrate neurons such as Purkinje and pyramidal cells. The mean-field model could be generalized by adding an addition class of branches that do not themselves branch and that spontaneously transition (catastrophe) to a long-lived shrinking state. The number would dependent on the branching rate and the length would be equal to the growth rate divided by the catastrophe rate. Embryonic *Drosophila* class I cells (Ferreira Castro et al., 2020b; Palavalli et al., 2021) and *C. elegans* PVD sensory cells (Heiman and Bülow, 2024) have a clear hierarchy of primary, secondary and tertiary branches."

2) Can this model be adapted to capture a two-phase process where primary branches are first established, and secondary branches subsequently emerge from these backbones? If the mean-field approach is inherently limited to treating all branches as equivalent, this limitation should be explicitly stated.

We believe that our model can be so adapted to capture a two-phase process ("step-step" branching) as described above (point 1).

3) If the authors believe their model does capture these experimental results, they should clearly explain how this happens and highlight the specific elements of the model that account for the "backbone-first" growth observed experimentally.

Our model can account for two-phase branching but not for more general hierarchical processes (we argue that class IV cells do not have hierarchical branching).

4) If this model represents an alternative explanation, the authors should provide a justification for why the current approach provides unique insights or is advantageous relative to prior work.

The unique insight is that the mean-field model does not have topology, so serves as a null hypothesis for hierarchical mechanisms, as argued in the Introduction.

Point-by-point

The authors' model assumes that the dendritic arbor grows as a 2D planar system, but this assumption is not sufficiently discussed. Since some biological neurons grow in 3D, it would be beneficial to discuss the potential limitations of this assumption and how it impacts the generalizability of the model. Additionally, the authors should highlight the extent to which their model can generalize to dendrites that are not constrained to 2D space.

We now say in the Discussion:

The present model and the agent-based model of (Shree et al. 2022) are inherently two-dimensional; direct cell-cell collisions take place when the growing dendrite and the target dendrite grow on a 2D surface, as is the case for class IV dendrites (Han et al., 2012). Collision is crucial because it provides negative feedback that keeps the dendrite density finite. In 3D, however, direct collisions will be much rarer because a growing line has a zero probability of colliding with a fixed line. While dendrites have non-zero thickness, the collision rates will still be very small, and a contact-based retract mechanism alone would lead to densities much higher than observed. Therefore, if a collision-based retraction mechanism were to operate in 3D, the effective size of the growing tip would need to be increased. This could be done by using filopodia, actin-based structures, to reach out from the growing tip and detect nearby dendrite branches. Filopodia are well known in the growth cones of axons, and also found in vertebrate dendrites (Fujishima et al., 2012; Portera-Cailliau et al., 2003) and adult fly dendrites (Yoong et al., 2020), though not in larval class IV dendrites. Morphogens that can diffuse through the tissue could also provide signaling cues that lead to retraction. Thus, contact-based retraction mechanisms might restrict dendrite density during the development of 3D arbors.

Finally, while the model focuses on class IV *Drosophila* dendrites, it is not clear if it can generalize to other neuronal types. The authors should explain how the model could be extended to other non-space-filling neuronal types (e.g., cortical pyramidal neurons) or address why it is restricted to space-filling morphologies.

In the Discussion, we now say:

“Our findings are expected to generalize to other arbors because the branching, elongation, and retraction mechanism (including contact-based retraction) has been observed in many other neuronal types (see references in the Introduction), though the specific parameters of dendrite growth will lead to cell-type dependent morphological variations (<https://neuromorpho.org>) Akram et al. (2018). However, there are important differences between class IV arbors and the arbors of other cells that restrict the general applicability of our model.”

In the next paragraph (in the last point), we say that contact-based retraction might generalize to 3D.

Data and Methodology:

The authors claim that the mean-field approximation offers clear advantages over branch-tracking models, which track the position, state, and hierarchy of individual dendritic branches. This is a compelling argument, especially in the context of the computational demands of tracking thousands of dynamic branches. However, with the rise of computer vision techniques that now allow automated branch tracking "for free", it is important for the authors to clearly articulate why the mean-field approach remains advantageous. The authors should highlight the computational, conceptual, or interpretive benefits of mean-field modeling that remain valuable even in the presence of automated tracking methods. This is key taking into account the previous work from

Point-by-point

the lab (Shree et al, 2022). At this point, it is not clear how the current manuscript differentiates itself from Shree and colleagues, 2022.

If the key advantage is computational efficiency, this should be quantified. If the advantage is conceptual (e.g., a more general view of dendritic growth dynamics), this should be clearly stated in the introduction or discussion.

We have extended the Introduction to articulate the differences and clear advantages of our mean-field model over our previous work.

1. it can test for topological models where branching and growth depend on the topology or hierarchy of branches in the dendrite network
2. it provides conceptual advantages as solutions can be obtained analytically.

The computational efficiency is not an issue as the Shree model can be solved on a personal computer.

The authors use a mean-field approach to describe the growth dynamics of dendritic tips. This approach simplifies the complexity of dendritic growth by focusing on average properties rather than individual branch states. However, it is critical to address the following issues:

1) Clarity of the model's assumptions: The assumption that the dendritic arbor grows as a 2D planar system is stated but not sufficiently discussed. Since most biological neurons grow in 3D, it would be beneficial to discuss the potential limitations of this assumption and how it impacts the generalizability of the model.

This is now included in the Discussion (see above).

2) Description of simulation procedures: There is reliance on simulations (both directed-rod and mean-field) without clear, detailed descriptions of the simulation procedure. Readers may be left wondering about the exact steps used to generate the data. Providing a clear, step-by-step explanation of how the simulations are performed, including key parameters, would address this issue.

We have now added a clear step-by-step simulation procedure (pseudo-code) in the Supplementary Information section.

3) Code availability: No code is provided to reproduce the figures. This limits the reproducibility and transparency of the study. Making the code available would enhance transparency, allow for validation of the model, and facilitate future work that builds on this model.

We have made the code available at https://github.com/SabyasachiSutradhar/Directed_Rod_Simulation.

4) Comparison with backbone-first models: Experimental studies, such as Yoong et al. (2020, Neuron) and Baltruschat et al. (2019, bioRxiv), propose a "backbone-first"

Point-by-point

process for dendritic growth. The authors should explain how their model accommodates, incorporates, or contradicts this process.

As argued, we do not think there is good evidence for “backbone-first” growth in class IV dendrites.

5) Biological Implications: The theoretical aspects of branching are well-covered, but the biological relevance is underexplored. For example, the authors should discuss how the model helps explain unique features of neuronal types beyond class IV *Drosophila* cells.

In the Introduction, we list many important biological features of neurons in general (not just *da* neurons), including highly branched morphologies, exponential length distributions, radial orientation, efficiency of space-filling. We show in the paper that all these biological features (and more) can be explained by the mean-field model. Thus, there are many biological implications.

We explain how biologically relevant features emerge from the model. For example, the exponential length distribution follows from the independence of collision on branch length. Also, the radial orientation follows from the competition between radially growing branches and non-radially growing branches. We make these points in the text.

6) Potential confounding factors: External factors like the impact of the extracellular matrix, or guidance cues are not fully explored. Addressing these confounding factors would provide a more comprehensive picture of dendritic morphogenesis.

In the Discussion, we have expanded on the limitations and caveats of our current model, including the role of external factors.

Suggested Improvements

1) Clarify the research question: The authors should articulate the specific problem or gap in knowledge that this study addresses earlier in the manuscript.

We direct the reviewer to the modified and expanded Introduction.

2) Highlight the model's novelty: Clearly define how the mean-field approach provides new insights that are not possible with existing models.

Likewise, this is articulated in the new Introduction.

3) Include simulation details: Provide a clear, step-by-step explanation of the directed-rod and mean-field simulation procedures, including all parameters.

We have now added a clear step-by-step simulation procedure in the Supplementary Information section.

Point-by-point

4) Address 2D vs 3D assumptions: Discuss the impact of the assumption that dendritic growth is constrained to 2D and its limitations.

In the revised Discussion.

5) Provide the code for reproducibility: Make the code publicly available to enable others to reproduce the figures and validate the findings.

We have made the code available at https://github.com/SabyasachiSutradhar/Directed_Rod_Simulation.

6) Address potential confounding factors: Discuss how the model might be influenced by extracellular matrix, or guidance cues, and clarify how these elements could be incorporated (or not) into the model.

Added to the revised Discussion.

7) Address backbone-first models: Discuss the model's relationship with the "backbone-first" growth process proposed by Yoong et al. (2020, Neuron) and Baltruschat et al. (2019, bioRxiv).

Please see previous answers on this topic.

8) Adjust text-to-equation ratio: Provide more explanatory text around the key equations, especially for readers unfamiliar with modelling

As suggested by the reviewer, we have added a section the results called "Three-state mean-field model" where we describe the model in non-mathematical terms.